# Phage parasites targeting phage homologous recombinases provide antiviral immunity

Gianluca Debiasi-Anders [1,2,4], Cuncun Qiao [1,2,4], Amrita Salim[1,2], Na Li[1,2] & Ignacio Mir-Sanchis [1,2,3] ✉

Bacteria often carry multiple genes encoding anti-phage defense systems, clustered in defense islands and phage satellites. Various unrelated anti-phage defense systems target phage-encoded homologous recombinases (HRs) through unclear mechanisms. Here, we show that the phage satellite SaPI2, which does not encode orthodox anti-phage defense systems, provides antiviral immunity mediated by Stl2, the SaPI2-encoded transcriptional repressor. Stl2 targets and inhibits phage-encoded HRs, including Sak and Sak4, two HRs from the Rad52-like and Rad51-like superfamilies. Remarkably, apo Stl2 forms a collar of dimers oligomerizing as closed rings and as filaments, mimicking the quaternary structure of its targets. Stl2 decorates both Sak rings and Sak4 filaments. The oligomerization of Stl2 as a collar of dimers is necessary for its inhibitory activity both in vitro and in vivo. Our results shed light on the mechanisms underlying antiviral immunity against phages carrying divergent HRs.

The bacterial immunity against viruses is clustered in defense islands and phage parasites[1–3]. Phage parasites (also referred to as satellites) are highly ubiquitous across the earth's biomass and significantly impact the ecology and evolution of bacteria in any microbiome on the planet[4]. A prototypical family of phage parasites is the *Staphylococcus aureus* Pathogenicity Islands (SaPIs). Traditionally, SaPIs have been seen as beneficial for the bacterial hosts due to their genetic cargo, usually associated with pathogenesis[5]. Their high prevalence has been attributed to their viral hijacking ability against the phage they parasitize. To parasitize a phage, the SaPIs employ a transcriptional repressor, Stl, which in addition to repressing the SaPI genes, senses an active phage infection by binding to the phage-encoded anti-repressor protein. The anti-repressor protein, here referred to as target, is SaPI specific. This way, SaPI2-encoded Stl (Stl2) targets the phage HRs, whereas SaPI1-encoded Stl targets a phage-encoded protein called Sri and SaPIbov1-encoded Stl targets the phage-encoded dUTPase[6] (Fig. 1A). However, these features do not fully explain the ubiquity of phage parasitizing mobile elements in natural environments.

Here we show that phage parasites such as SaPIs provide robust antiviral immunity without carrying an anti-phage system. Such immunity is mediated by Stl, the transcriptional repressor. Among the three prototypical SaPIs tested, we found that SaPI2 provided the highest level of immunity. SaPI2-encoded transcriptional repressor, Stl2, targets the phage-encoded HR as anti-repressor[7]. Phylogenetic and modeling studies have classified phage recombinases into three superfamilies: Rad51-like, Rad52-like, and Gp2.5-like[8–10]. Stl2 targets HRs spanning all superfamilies. Therefore, it is an outstanding question how Stl2 might target and inhibit HRs adopting highly divergent oligomeric architectures such as rings (Rad52[11]) and filaments (Rad51[12]). Here we have focused on two recombinases, an annealase called Sak, and a recombinase called Sak4, belonging to the Rad52-like and Rad51-like superfamilies respectively. To reflect the phages that harbor them, we refer to these recombinases as Sak$_{80\alpha}$ and Sak4$_{52A}$. Note that we use the term annealase for Rad52-like proteins and recombinases (HRs) for Rad51-like proteins or when referring to both types collectively. Given the importance of homologous

[1]Department of Medical Biochemistry and Biophysics, Umeå University, Umeå, Sweden. [2]Wallenberg Centre for Molecular Medicine, Umeå, Sweden. [3]Institute for Bioengineering of Catalonia (IBEC), The Barcelona Institute of Science and Technology (BIST), Baldiri i Reixac 10-12, Barcelona, Spain. [4]These authors contributed equally: Gianluca Debiasi-Anders, Cuncun Qiao. ✉e-mail: ignacio.mir-sanchis@umu.se

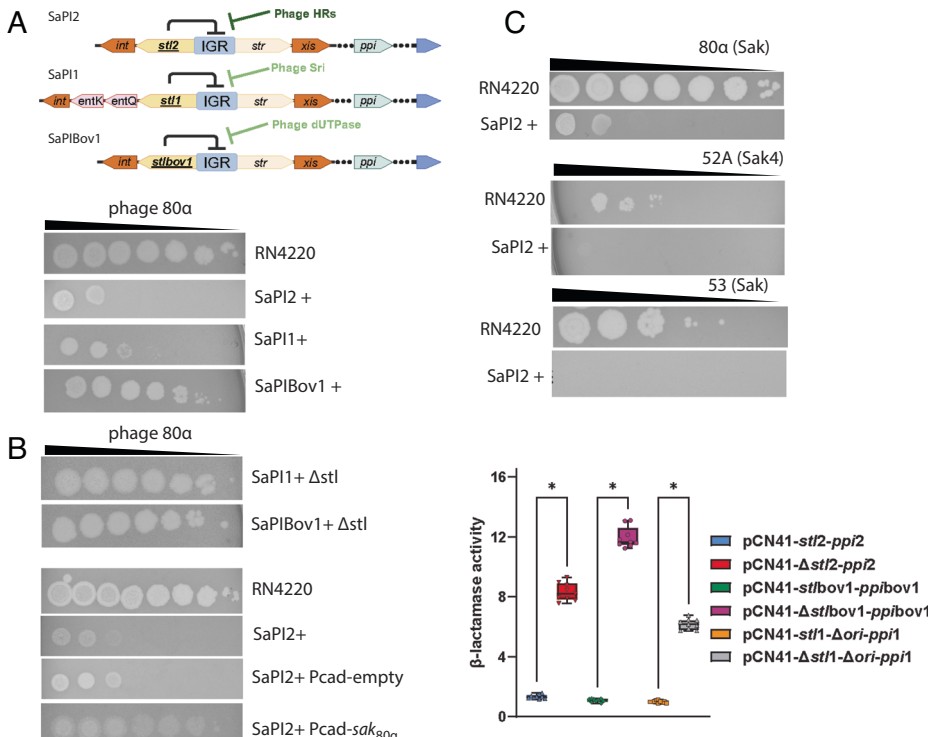

**Fig. 1 | SaPI-encoded Stls mediate robust antiviral protection. A** Top, cartoon exemplifying the regulation-induction processes of SaPI2 (top), SaPI1 (middle), and SaPIbov1 (bottom). The arrows represent gene orientation. IGR, intergenic region between the leftward and rightward operons; *int*, integrase; *ent*K and *ent*Q, enterotoxins K and Q respectively; *stl*, SaPI transcriptional repressor left; *str*, SaPI transcriptional regulator right; *xis*, excisionase; *ppi*, phage packaging interference. Phage encoded HRs, Sri, and dUTPase are the anti-repressors (referred to as targets throughout the manuscript) for SaPI2, SaPI1, and SaPIbov1 respectively. Bottom gels, SaPIs provide robust protection against phage 80α. Spot assays where tenfold serial dilutions of phage 80α lysates were spotted on lawns of the RN4220 strain harboring no SaPI or SaPI2, SaPI1, and SaPIBov1. **B** SaPI-encoded Stls mediate antiviral protection. Left, first two top panels, the same phage lysate dilutions as in A spotted on lawns of RN4220 strain carrying SaPI1Δstl and SaPIBov1Δstl. In the four bottom panels, serial dilutions of phage 80α lysates were spotted on RN4220, RN4220 carrying SaPI2 (SaPI2+), RN4220 SaPI2+ carrying empty pCN51-derivative plasmid pGTM3 (SaPI2+ Pcad empty) and RN4220 SaPI2+ expressing the *sak*$_{80α}$ gene (SaPI2+ Pcad-*sak*80α). On the right-hand side, the box plot shows the nitrocefin-based transcriptional fusion assays where the β-lactamase reporter gene

of the pCN41 plasmid was fused to the *ppi* genes of SaPI2, SaPIbov1, and SaPI1 in the presence or absence of their cognate *stl* gene. Note that the SaPI1 constructs lack the origin of replication, which is upstream of the *ppi* gene, see methods. β-lactamase activity is defined as increments in absorbance over time in minutes. See methods. Data are presented as individual dots on top of the box-whisker-plot where the whiskers represent the minimum and the maximum values, the borders of the box represent the 25th and the 75th percentile, while the horizontal line crossing the box represents the median. (*) $P$ value < 0.05 was considered statistically significant. Three independent biological replicates ($n = 3$) with each having three experimental replicates were used in one way ANOVA followed by post hoc for multiple comparisons statistical test calculated in GraphPad giving a $P$ value < 0,000001. **C** SaPI2 provides robust antiviral protection against phages encoding HRs from different superfamilies. Spot assays as in (**A**), where the tenfold dilutions of lysates from phages 80α, 52A, and 53 were spotted on lawns of the RN4220 strain with and without SaPI2. The acronyms of the different phage HRs are shown in parentheses. In all cases (**A**–**C**) the assays were repeated at least three times. Source data are provided as a Source Data file.

recombination, it is not surprising that several lines of evidence indicate that unrelated phage defense systems target phage-encoded HRs spanning all superfamilies. The study of plasmid-encoded Abortive infection mechanisms (Abi) led to the identification and initial characterization of proteins in *Lactococcus lactis* bacteriophages named Sak (Sensitivity to AbiK)[13]. A few years later it was discovered that an Abi system in *Staphylococcus aureus* also targets phage recombinases[14]. Recently, CRISPR spacers have been found matching single strand annealing protein genes (*ssap* genes) in Firmicutes indicating that CRISPR systems also target phage annealases[9].

We present the Stl2-mediated inhibitory mechanism against Sak$_{80α}$ (Rad52-like) and Sak4$_{52A}$ (Rad51-like) through a comprehensive analysis of cryo-EM structures and accompanying experiments. We reveal how Stl2 inhibits phage HRs folding in such divergent architectures. Our examination of the cryo-EM structure of apo Stl2 shows its tendency to form both ringed-shaped and filamentous oligomers of dimers, indicating that Stl2 mimics the oligomeric state of its targets. We provide the cryo-EM structures of a truncated version of Sak$_{80α}$ and two Stl2-target complexes: Stl2-Sak$_{80α}$ and Stl2-Sak4$_{52A}$-ATPγS complexes, both in the mega Dalton range. Stl2 decorates both targets,

generating an impressive architectural scaffold. We establish that key residues mediating Stl2's oligomeric state are likewise crucial for providing antiviral immunity. This finding underscores the evolution of Stl2's ability to oligomerize into large multimers as essential for targeting structurally divergent phage recombinases. Our work might open new avenues for designing inhibitors of homologous recombinases as biomedical tools.

## Results

### SaPI2 provides robust protection against phages encoding different homologous recombinases

In planktonic experiments, SaPIs are known to reduce phage burst size by 10-100 fold[5,15]. The effect that SaPIs might have during phage infection has not been studied using spot assays. Therefore, we investigated whether a SaPI-harboring *Staphylococcus aureus* strain could protect against staphylococcal phages in spot assays (Fig. 1A). We used RN4220 strains with three prototypical SaPIs (SaPI1, SaPI2, and SaPIbov1) challenged with phage 80α, known to encode for anti-repressors of these SaPIs. SaPI1 and SaPI2 islands provided a robust protection, while SaPIbov1 offered mild protection. SaPI2-mediated

protection surpassed SaPI1. We hypothesized that the varying degrees of protection observed with these SaPIs might be attributed to their Stls targeting different phage-encoded anti-repressors or, alternatively, to the presence of uncharacterized mechanisms related to phage immunity within the SaPIs. To investigate this, we utilized variants of SaPI1 and SaPIbov1 in which the *stl* genes had been inactivated through genomic mutagenesis: SaPI1Δ*stl*, SaPIbov1Δ*stl*[16]. In these variants, it is generally presumed that all SaPI genes are constitutively expressed, leading us to hypothesize that any characterized or uncharacterized factor encoded by the SaPIs would be in place to interfere with phage reproduction during the spot assays. Surprisingly, we discovered that the absence of the *stl* gene in SaPI1Δ*stl* and SaPIbov1Δ*stl* variants turned them sensitive to phage infection (Fig. 1B). We then turned to SaPI2. After many attempts, we failed to generate a *stl*2 mutant. To overcome this limitation, we performed an experiment where we tried to mimic the *stl*2 mutant genotype. In this experiment, we cloned the phage 80α-encoded annealase $sak_{80\alpha}$ in the expression vector pGTM3, which is a pCU1 derivative carrying the cadmium inducible promoter Pcad[17,18]. We then transformed the construct to the RN4220 strain harboring SaPI. After plasmid induction with cadmium, $Sak_{80\alpha}$ should interact with Stl2 and the SaPI2 island would be derepressed[7], mimicking a *stl*2 deficient state. We hypothesized that performing spot assays with phage 80α under these conditions would simulate a likewise scenario as with SaPI1 Δ*stl* and SaPIbov1 Δ*stl*, in which the immunity provided by SaPI2 would be lost because Stl2 would be sequestered by the ectopically expressed $Sak_{80\alpha}$. As shown in Fig. 1B this was the case.

SaPI2 as well as SaPI1 and SaPIbov1 encode for two interference mechanisms against phage 80α, named *ppi* (phage packaging interference) and *cpm*AB (small capsids morphogenesis)[19–21]. It is assumed that these mechanisms should be expressed in a *stl* mutant (or after anti-repressor ectopic overexpression). To fully confirm this assumption, we measured the expression of *ppi* in SaPI2, SaPIbov1 and SaPI1. Using the well-established pCN41 plasmid we used transcriptional fusion assays[7,17,22] to measure the expression of the *ppi* genes in the presence or absence of their cognate *stl*. Note that for the case of the SaPI1 constructs, we had to delete the SaPI-containing origin of replication (Δ*ori*), since leaving the *ori* intact seemed to interfere with the cloning process. As shown in Fig. 1B, within the three islands, the absence of *stl* implied the expression of the cognate *ppi*. These experiments indicated that a physiological constitutive expression of SaPI-encoded genes in *stl* deficient variants was not sufficient for phage protection. Moreover, these results suggested the necessity of *stl* for full immunity (Fig. 1B). We then asked if *stl* was sufficient to provide phage protection against phage 80α. We cloned the SaPI2-encoded *stl*2 gene into the expression vector pCN51 and transformed to RN4220 strain. After *stl*2 ectopically overexpression, we infected the cells with phage 80α. We discovered that cells were sensitive to phage infection (not shown), indicating that Stl2 was not sufficient to provide immunity against phage 80α in these experimental conditions.

We then focused on SaPI2 and its immunogenic effect against various phages. We included phages carrying HRs belonging to two different superfamilies. Phages 80α and 53 encode for Sak, a Rad52-like annealase, while phage 52A encodes for Sak4, belonging to the Rad51-like superfamily (Fig. 1C). Despite phage variability, SaPI2 provided robust immunity against all tested phages (Fig. 1C). These findings suggest that SaPIs, exemplified by SaPI2, and similar phage parasites are much more effective at inhibiting phage physiology than previously thought. Altogether, our experiments suggested that these SaPIs do not carry uncharacterized anti-phage systems and that *stls* are a necessary condition in terms of immunity, albeit not a sufficient condition when overexpressed from a plasmid. In our view, SaPIs offer a unique model for studying anti-phage defense in a physiological scenario. We hypothesized that SaPI2's superior protection was due to

Stl2 targeting and inhibiting phage HRs, prompting further characterization of this mechanism.

## Stl2 as transcriptional regulator

We expressed and purified the carboxy terminally histidine tagged full-length Stl2 using *E. coli*. Its activity as a transcriptional regulator was first confirmed through electromobility shift assays (EMSAs), indicating its capability to bind the intergenic region (IGR) between *stl* and *str* divergent genes (Fig. 2A). A subsequent DNAse-I footprinting assay revealed protection of two direct repeats, denoted left and right, of 18 nucleotides (nt), with a small flanking patch also protected (Supplementary Fig. 1). Modifying the repeats' sequence confirmed them as Stl2 operators (Supplementary Fig. 1).

During the Stl2 purification procedure, where we assumed Stl2 as a dimer given its role as transcriptional regulator, we observed that Stl2 eluted prematurely in the superose column (not shown). Investigating Stl2's oligomeric state by mass photometry showed two peaks, one corresponding to the dimeric form (54 kDa) and a second peak of 626 kDa, indicating that Stl2 displayed distinct oligomeric states in solution (Fig. 2B). To further analyze the oligomeric state of Stl2 in solution we solved the apo Stl2 structure by cryo-EM.

## Stl2 mimics the architecture of its targets

Surprisingly, the micrographs showed circular structures as well as long curved filaments (Fig. 2C), highlighting Stl2's intrinsic tendency to oligomerize mimicking its target's architecture. Extracting particles with box sizes sufficiently large to enclose the circles allowed us to identify ring-shape structures of 11, 12, and 13 Stl2 protomers by inspecting the 2D classes. We generated a cryo-EM map of the undecamer state with C11 symmetry and discovered that the circles were composed by 11, 12, and 13 dimers of Stl2. Such dimers were formed in the dihedral axis. We then generated an initial map and model imposing D11 symmetry (Supplementary Fig. 1F). We extracted particles with a smaller box size so that three complete Stl2 dimers were resolved imposing D1 symmetry at an overall 3.7 Å resolution (Fig. 2C, D, Supplementary Fig. 2, Supplementary Table 1). Notably, at the interfaces of monomer-monomer and dimer-dimer contacts, the local resolution improved to 2.5 Å (Supplementary Fig. 2). The structure of the dimer displays a peculiar butterfly shape, where the N-terminal domain (NTD) and the C-terminal domain (CTD) are the body and the wings of the butterfly respectively (Fig. 2C, D). Truncated versions confirmed that the NTD, harboring a winged helix turn helix motif (wHTH), is responsible for DNA binding activity in vitro, contrary to the CTD, whose deletion caused a 30-fold reduction in Stl2's binding affinity for the DNA compared to the wild type (Fig. 2E, Supplementary Fig. 1). We then asked if the ability to bind the DNA was coupled with the ability of the Stl2 dimers to oligomerize as circles/filaments. The dimer-dimer interface corresponds to a patch of the antiparallel β-sheet present in the CTD and the α-helix 2 present in the NTD (Fig. 2D). The α-helix 2 is part of the wHTH motif, enriched with positively charged residues and most likely playing a role in DNA binding (Fig. 2D, Supplementary Fig. 1F). With the intention to disrupt the dimer-dimer interaction we generated two mutants. We called these mutants Stl2-helixNTD and Stl2-sheetCTD. Within the Stl2-helixNTD mutant, the residues K16, R19, K22, present in α-helix 2 and R28 in α-helix 3 were substituted with alanine residues. In the case of the Stl2-sheetCTD mutant, the residues E106, L108, D110, K118, N120 and D122 present in the antiparallel β-sheet located in the CTD were mutated to alanine residues (Supplementary Fig. 2E). We first analyzed these two mutants by mass photometry and confirmed that they displayed only one peak corresponding to the dimeric state, indicating that they had lost their ability to form circles/filaments of dimers (Supplementary Fig. 1C). We then performed the EMSAs and discovered that Stl2-helixNTD had lost its ability to bind the DNA contrary to the Stl2-sheetCTD mutant which was still active regarding its DNA binding activity (Fig. 2E,

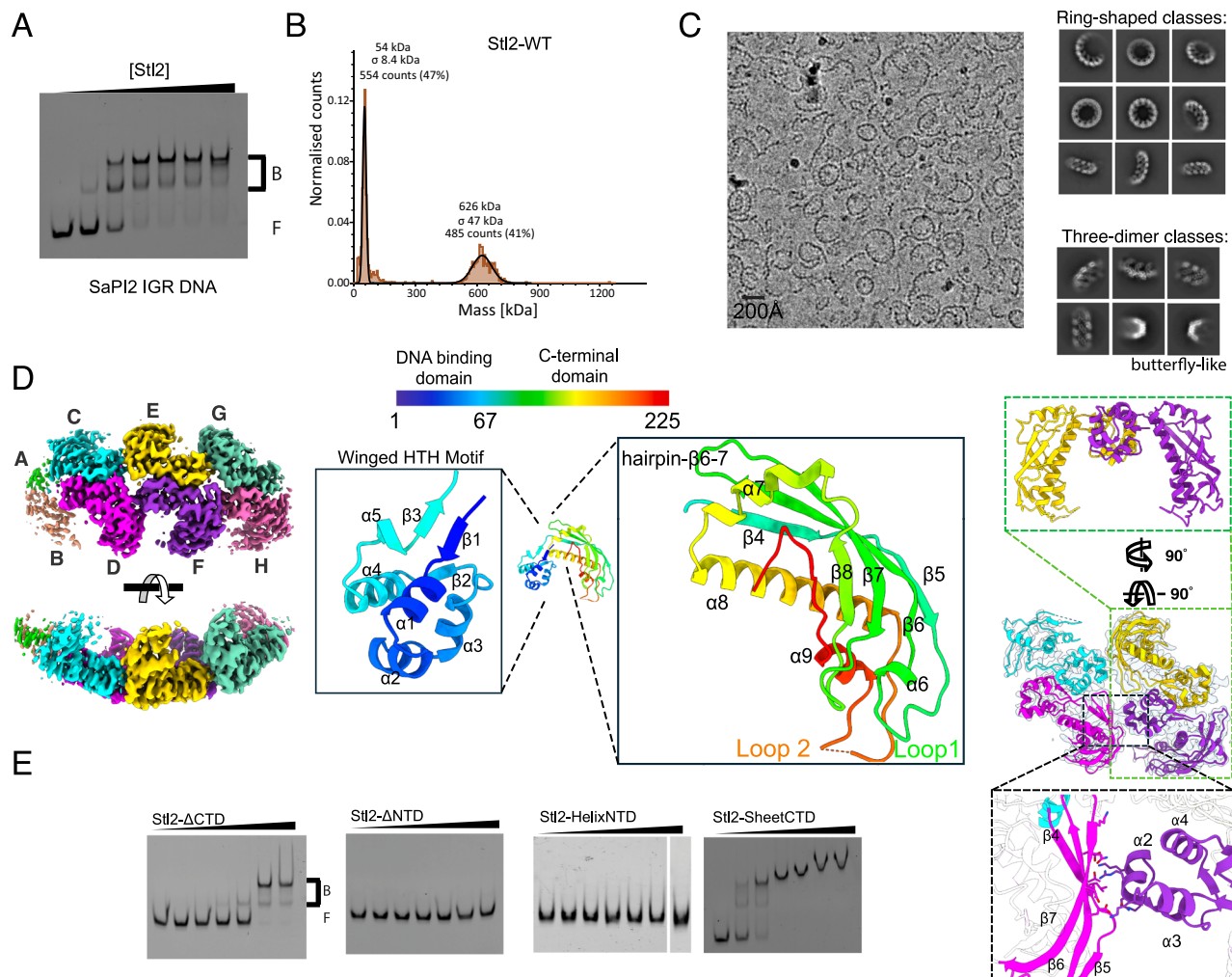

**Fig. 2 | Characterization of Stl2. A** Stl2 binds to SaPI2 IGR. EMSA of wild-type Stl2 with SaPI2 IGR. The 212 base pairs in the SaPI2 IGR were FAM-labeled PCR-amplified and used as substrate at 2 nM final concentration. Monomeric Stl2 concentrations were 0, 1, 5, 10, 50, 100, 500 nM. F, free DNA, B, bound DNA. A representative gel of three independent biological replicates is shown. **B** Apo Stl2 displays two distinct oligomeric states in solution. Normalized mass photometry plot of apo Stl2 wildtype. Above each peak, the mass (in kDa), the sigma value (in kDa), and the number and percentage of counts are shown. **C** Stl2 oligomerizes as circles and filaments. Left, a representative cryo electron micrograph out of the 6233 collected, showing Stl2 particles adopting circular shapes as well as open filaments. The scale bar is 200 Å. Right, the 2D class averages of particles are separated into 2 groups: top, averaged particles extracted with large box size (500 Å) to cover closed and open circles and labeled ring-shaped classes; bottom, particles with smaller box size (300 Å) to crop three dimers, which were subsequently used for the final 3D reconstruction. The bottom right corner shows two averages of the butterfly-like shape. **D** Cryo-EM structure of Apo Stl2. Left, orthogonal views of the cryo-EM map. Densities for four dimers were visible, but only three dimers (C, D cyan-pink, E, F yellow-purple, G, H green-pale red) were

used for model fitting. Middle top, domain organization of Stl2 protomer depicted as rainbow-colored bar. The numbers reflect amino acid residues. Middle bottom, rainbow ribbon representation of the Stl2 protomer domains, where α-helices and β-strands are consecutively numbered from N (blue) to C (red). Right, dimer-dimer stabilizing interactions. The middle panel shows two dimers: (C, D) cyan-pink, and (E, F) yellow-purple. Two zoomed regions are shown: top, two consecutive -90° rotations in the horizontal and vertical axes evidence the yellow-purple dimer to show the characteristic butterfly-like shape of the dimer; bottom, zoomed area of the dimer-dimer contacts between the pink and purple protomers. The residues whose side chains are shown as sticks are K16, S18, R19, K22, R28 in the purple protomer's α-helix 2 and T72, E106, L108, D110, K118, N120, and D122 in the antiparallel β-sheet present in the pink's protomer CTD. **E** Structure-based mutagenesis analysis of Stl2. EMSAs with same DNA concentration as in A with various Stl2 mutants whose monomeric concentrations were 0, 5, 10, 50, 100, 500, and 1000 nM for the truncates and the helixNTD mutant, whereas the sheetCTD concentration gradient was as the Wild-type in (**A**). F, free DNA, B, bound DNA. Representative gels of three replicates are shown. See text and Supplementary Fig. 1. Source data are provided as a Source Data file.

Supplementary Fig. 1D). These data confirmed that the α-helix present in the NTD of Stl2 harbors key residues for its DNA binding and oligomerization activities. Moreover, the findings from the Stl2-sheetCTD mutant suggest that Stl2's capacity to oligomerize into circles/filaments is independent of its DNA-binding ability.

To complement our in vitro data with a more physiological in vivo scenario, we used transcriptional fusion assays[7,22]. Taking advantage of the well-established pCN41 plasmid[17], we cloned the SaPI2-containing regulatory region comprising from the *stl2* gene to the middle of the

*xis* gene, to which the β-lactamase gene was fused (Supplementary Fig. 1E). This wildtype construct mimics the quiescent state of the SaPI2 island, where Stl2 binds to its operators and the β-lactamase gene is repressed. We then tested in vivo all the mutations designed earlier in vitro (Supplementary Fig. 1E). Consistent with the previous EMSA results, the *stl2*-ΔNTD truncate and the *stl2*-helixNTD mutant that were unable to bind the cognate DNA in vitro were also unable to act as transcriptional repressors in vivo. Interestingly, the *stl2*-ΔCTD truncation also failed to act as a repressor, implying that a 30-fold reduction

in DNA binding affinity measured by the EMSAs signifies a null activity as a transcriptional repressor in vivo (Supplementary Fig. 1D, E). Regarding the *stl2*-sheetCTD mutant, we discovered that it was able to act as transcriptional repressor which is in agreement with the previous EMSAs (Fig. 2E, Supplementary Fig. 1D, E). Since the formation of Stl2 circles/filaments was found to be unrelated to its function as a transcriptional repressor both in vitro and in vivo, we hypothesized that it might be associated with its activity against phage-encoded HRs. We, therefore, sought to characterize Stl2's inhibition activity against the phage 80α annealase, $Sak_{80\alpha}$.

## $Sak_{80\alpha}\Delta CTD$ is a human RAD52 homolog

We struggled to obtain a good quality cryo-EM map of full-length $Sak_{80\alpha}$. The full-length dataset presented two major challenges: severe particle orientation preference and the absence of the flexible CTD, which is a common issue with Rad52-like annealases. Consequently, we decided to focus on a truncated version of $Sak_{80\alpha}$ where its CTD was removed, $Sak_{80\alpha}\Delta CTD$. The data set of $Sak_{80\alpha}\Delta CTD$ exhibited improved behavior. The 2D averages from this dataset revealed stable ringed structures of different sizes, with octadecamers being the predominant form, which was used for the final reconstruction (Fig. 3A, Supplementary Fig. 3, Supplementary Fig. 4, and Supplementary Table 1). The $Sak_{80\alpha}\Delta CTD$ truncate version folds similarly to other SSAPs[11,23–25]. It shows the groove commonly seen in SSAPs to accommodate ssDNA in the outer surface of the torus, known as the ssDNA binding groove or ssDBG (Fig. 3B, Supplementary Fig. 3A). Structural comparisons performed by Dali[26] indicated that the closest homolog is the human RAD52 (5JRB) protein with a Z-score of 8.2, followed by the coliphage lambda annealase Redβ (7UJL, Z-score 5.7). $Sak_{80\alpha}\Delta CTD$'s most notable difference within this domain is a long insertion in the β-strand 3 which potentially obliges the adjacent protomer to slightly rotate compared to the human protein (Fig. 3B).

To fully characterize $Sak_{80\alpha}$ as an annealase, we first tested that $Sak_{80\alpha}$ was able to bind ssDNA but not dsDNA (Supplementary Fig. 3D).

We then measured $Sak_{80\alpha}$'s ability to anneal complementary oligonucleotides by annealing assays (Fig. 3C, Supplementary Fig. 3B). Peak activity (95%) was observed at 10 nM ring concentration to which point the activity gradually decreased and plateaued to around 75–80%. Since it is well known that SSAPs coordinate with single strand binding proteins (SSB), we included the phage 80α SSB ($SSB_{80\alpha}$). $SSB_{80\alpha}$ demonstrated its capability to bind ssDNA but not dsDNA (Supplementary Fig. 3E). We then verified that $SSB_{80\alpha}$ did not show annealing activity per se and confirmed that it prevented spontaneous annealing of the complementary oligonucleotide (Fig. 3D, Supplementary Fig. 3B). We repeated the annealing assay this time including both $Sak_{80\alpha}$ and $SSB_{80\alpha}$-ssDNA complex into the reaction mix thus closer to physiological conditions. The presence of $SSB_{80\alpha}$ decreased product formation at lower concentrations and elevated the plateau at higher concentrations, describing a hyperbolic curve (Fig. 3E, Supplementary Fig. 3C). This experiment suggested that $Sak_{80\alpha}$ was able to interact with $SSB_{80\alpha}$-ssDNA complexes, displaced $SSB_{80\alpha}$ and annealed complementary oligonucleotides, thus indicating that both proteins work in concert. Truncated version of each protein where their CTDs were removed ($Sak_{80\alpha}\Delta CTD$, $SSB_{80\alpha}\Delta C7$), proved that these two proteins coordinate their functions via their CTDs (Fig. 3E, Supplementary Fig. 3C).

## Mutual inhibition between Stl2 and $Sak_{80\alpha}$

From a physiological perspective, the presence of the phage-encoded HR hinders Stl2 from binding to its cognate operators[6,7]. To test this, we conducted a new series of EMSAs including increasing amounts of $Sak_{80\alpha}$ in the reaction mix and maintaining constant the DNA and Stl2 concentrations. $Sak_{80\alpha}$ prevented Stl2 from binding to its cognate DNA (Fig. 4A). We then accompanied these data with the in vivo transcriptional fusion experiments. For this purpose, we used the previous construct where we cloned the *sak*$_{80\alpha}$ gene into the plasmid pGTM3[18]. We then co-transformed the pGMT3-*sak*$_{80\alpha}$-containing RN4220 strain with the pCN41-WT (wildtype) construct whose β-lactamase reporter gene was repressed when analyzed individually (Supplementary

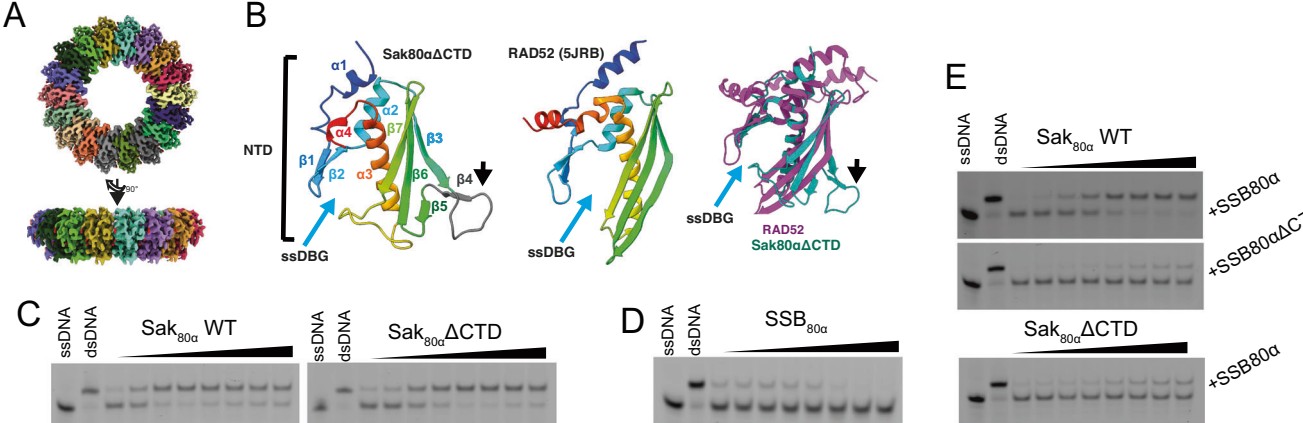

**Fig. 3 | $Sak_{80\alpha}$ is an annealase folding similar to the human RAD52. A** Map of the $Sak_{80\alpha}\Delta CTD$ truncate. Orthogonal views of the C18 symmetry-imposed map at 3.21 Å. The density is colored according to each protomeric model in Supplementary Fig. 3 A. **B** The model of Sak80αΔCTD folds like human RAD52. Left, $Sak_{80\alpha}\Delta CTD$ model colored as rainbow, α-helices, and β-strands are consecutively numbered from N (blue) to C (red). The insertion in β-strand 3 is shown in grey. The cyan and black arrows point to the ssDBG and the β-strand insertion observed in $Sak_{80\alpha}$ respectively. Middle, chain D of the RAD52 model PDBID 5JRB with residues 25-33 and 177-208 hidden and rainbow-colored. Right, the same models but overlayed and colored purple and green. ssDBG, ssDNA binding grove. Cyan and black arrows as in the left panel. **C** Sak80α is an annealase. Representative gels of annealing assays with wild-type $Sak_{80\alpha}$ and the $Sak_{80\alpha}\Delta CTD$ truncated version. ssDNA and dsDNA correspond to unreacted FAM-oligo26 and pre-annealed oligo-25/FAM-oligo-26 respectively; protein concentrations are 0, 1, 5, 10, 25, 50, 75, 100 nM calculated as heptadecamers. **D** $SSB_{80\alpha}$ inhibits spontaneous annealing between complementary oligos. ssDNA and dsDNA as in (**C**); protein concentrations are 0, 10, 25, 50, 75, 100, 150, 200 nM $SSB_{80\alpha}$ (tetrameric). **E** The $Sak_{80\alpha}$ CTD and $SSB_{80\alpha}$ C-terminal tail are both necessary for efficient annealing. Top gel, annealing assay between wild-type $Sak_{80\alpha}$ and wild-type $SSB_{80\alpha}$. Middle gel, reaction with wild-type $Sak_{80\alpha}$ and $SSB_{80\alpha}\Delta C7$ (truncated version where the last 7 residues were removed). Bottom gel, reaction with $Sak_{80\alpha}\Delta CTD$ and wild-type $SSB_{80\alpha}$. In all three gels, pre-incubations were done with the corresponding $SSB_{80\alpha}$ at 200 nM (tetrameric), while the corresponding $Sak_{80\alpha}$ gradient was 0, 1, 5, 10, 25, 50, 75, 100 nM (heptadecameric). The gels shown in (**C**–**E**) are representative of three independent replicates. Source data are provided as a Source Data file.

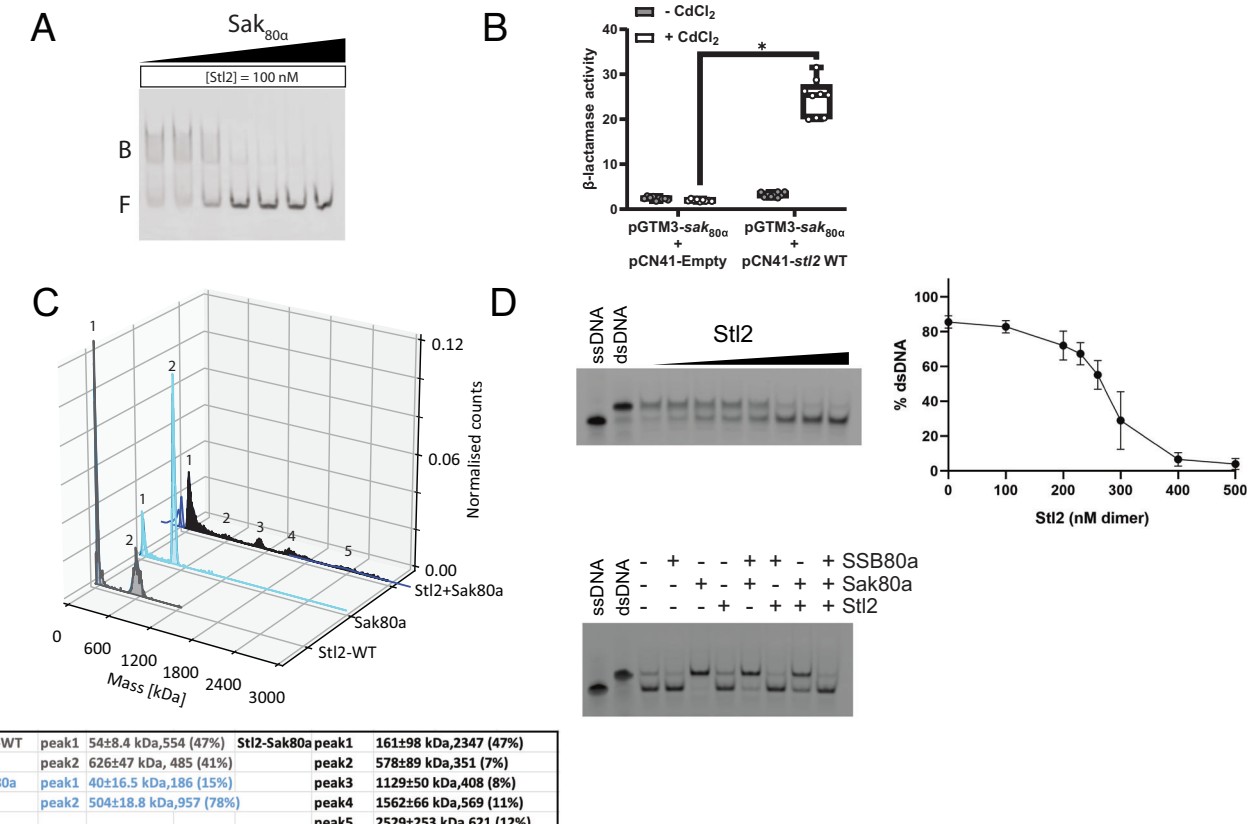

**Fig. 4 | Mutual inhibition between Stl2 and Sak80α. A** Sak80α inhibits Stl2's DNA binding activity in vitro. EMSA where a constant concentration of 100 nM of monomeric Stl2 was mixed with 2 nM DNA substrate and increasing amounts of Sak80α whose concentrations were 0, 50, 100, 200, 400, 600, and 1000 nM. The gel shown is representative of three independent replicates. **B** Sak80α inhibits Stl2's activity as transcriptional repressor. Nitrocefin-based β-lactamase in vivo assay of the RN4220 strain co-transformed with two plasmids: the pCN41 empty and wild-type versions used in Supplementary Fig. 1E plus the cadmium-inducible vector pGTM3 carrying the $sak_{80\alpha}$ gene. The β-lactamase activity was measured after the addition of none or 5 μM $CdCl_2$ to induce $sak_{80\alpha}$ expression. β-lactamase activity is defined as increments in absorbance over time in minutes. See methods. Data are presented as individual dots on top of the box-whisker-plot where the whiskers represent the minimum and the maximum values, the borders of the box represent the 25th and the 75th percentile, while the horizontal line crossing the box

represents the median. (*) $P$ value < 0.05 was considered statistically significant. Three independent biological replicates ($n = 3$) with each having three experimental replicates were used in an unpaired two-tailed $t$-test performed in GraphPad giving a $P$ value = 2.92 E-5. **C**, Corresponding mass distributions of the complex Stl2-Sak80α as per mass photometry. Mass (kDa), sigma (kDa), number of particles, and percentage are depicted below the graph. **D** Stl2 inhibits the DNA annealing reaction mediated by Sak80α and SSB80α. Top gel, 50 nM Sak80α (heptadecameric) was pre-incubated with a gradient of Stl2 and then added to the same SSB80α -oligo-25 reaction system as in Fig. 3E wild-types. The Stl2 gradient was 0, 100, 200, 230, 260, 300, 400, 500 nM of Stl2 (dimeric). The graph on the right-hand side represents the quantification of three gels. Points represent the mean, bars the sd. Bottom gel, permutated reactions between 200 nM of SSB80α (tetrameric), 50 nM Sak80α (heptadecameric), and 500 nM Stl2 (dimeric). Source data are provided as a Source Data file.

Fig. 1E). We then measured the expression of the β-lactamase gene with and without cadmium over-expression of pGTM3-$sak_{80\alpha}$. In agreement with previous literature[7] and our EMSAs, the over-expression of Sak80α with cadmium-induced Stl2 and prevented it from repressing the SaPI2 genes in vivo (Fig. 4B).

Although the artificial overexpression of $stl2$ in vivo was not sufficient to prevent phage reproduction under these experimental conditions, we sought to determine whether Stl2 would inhibit Sak80α in vitro. To test this, we first confirmed that Stl2 and Sak80α complexed together by analyzing them by mass photometry (Fig. 4C). We first ran separately Stl2 and Sak80α in the mass photometer. Similarly to Stl2, Sak80α showed two peaks corresponding to 40 kDa and 504 kDa. When both proteins were run together, the larger peaks decreased the number of counts while new counts appeared at masses corresponding to 1.1, 1.5, and 2.5 MDa, indicating heterogeneous complex formation between Stl2 and Sak80α.

To demonstrate that Stl2 could inhibit Sak80α, we repeated the annealing assays with the inclusion of Sak80α, SSB80α and a gradient of Stl2, which revealed the absence of annealed oligonucleotides at approximately 1:1 molar ratio (monomeric), indicating that Stl2

effectively inhibited Sak80α-mediated process (Fig. 4D). A comprehensive set of assays varying the presence of Sak80α, SSB80α, and Stl2 confirmed that Stl2 caused less inhibition when SSB80α was absent, compared to the greater inhibition observed when both phage proteins Sak80α and SSB80α were included in the reaction (Fig. 4D). Altogether, our findings demonstrate that Sak80α and SSB80α mutually facilitate their functions via their CTDs and that Stl2 inhibits Sak80α-mediated annealing activity. Importantly, our work marks the first evidence of such inhibition reaction.

## The structure of the Stl2-Sak80α complex reveals the Stl2-mediated inhibition of Sak80α

The micrographs exhibited significant particle heterogeneity (Fig. 5A, Supplementary Fig. 5). The most abundant particles revealed an architectural structure formed by fifteen Stl2 dimers acting as a molecular filigree decorating two heptadecameric Sak80α rings, keeping them apart and generating a ~1.5 MDa structure (Fig. 5B–D, Supplementary Fig. 5, Supplementary Table 1). As a result of the asymmetry between the number of Sak80α and Stl2 protomers, two Sak80α subunits in each ring were missing any CTD density while a

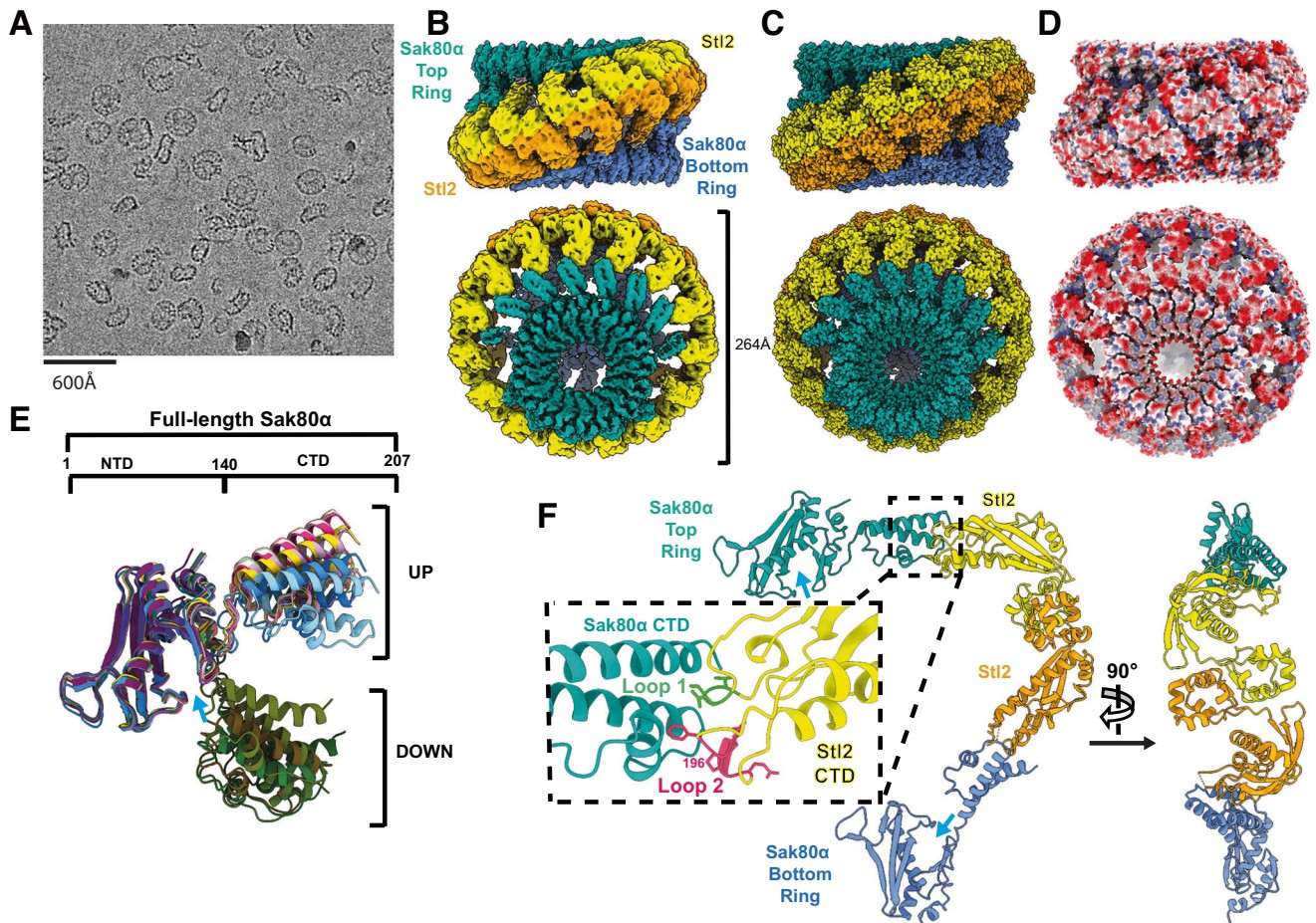

**Fig. 5 | The structure of full-length Sak80α in complex with Stl2 reveals the inhibition mechanism. A** Lowpass-filtered sample micrograph at 165000x magnification, presentative of the 27532 collected micrographs. Orthogonal views of the map (**B**) and model (**C**) shown as spheres. The full map consists of two heptadecameric Sak80α rings which are colored green (labeled top ring) and blue (bottom ring). Stl2 dimers are colored yellow and orange. **D** Electrostatic representation of the model in (**C**). **E** The Sak80α CTD is held in two conformations by its interaction with Stl2. The top diagram shows the number of the residues delimiting the Sak80α NTD and the CTD. The Sak80α protomers in one ring are overlayed at the NTD and

shown in different colors. The CTDs adopt two positional variations which are labeled UP (open) or DOWN (closed). The cyan arrow points to the ssDBG. **F** Segment of the model shown in C. The right image is the orthogonal view. The chains shown here are 1B (Sak80α, green), 2B (Stl2, yellow), 3L (Stl2, orange) and 4L (Sak80α, blue). See Supplementary Fig. 5 for chain identifiers. Cyan arrows mark the ssDBG. Inlet, magnified view of the interaction between Sak80α top ring and Stl2, which takes place via the CTD of both proteins. Stl2 amino acid side chains of residues Glu89, Val90, Thr91 and Phe195, Pro196, Asn197, Lys198, Glu200 are shown as sticks in olive green (loop 1) and pink (loop 2) respectively.

further two had at least partial albeit extremely poor-quality densities, which were left unmodelled (Supplementary Fig. 5).

The structure instantaneously elucidates how Stl2 is inhibiting Sak80α by interacting with and corsetting Sak80α's CTD (Fig. 5F). To our knowledge, this is the first visualization of the α-helical bundle present in the CTD of a full-length Rad52-like annealase. An overlay of Sak80α models revealed that the CTD was fixed in two different conformations, either up or down (Fig. 5E), which in relation to Sak80α's ssDBG might be seen as open (up) or closed (down). This flexibility in Sak80α, provided by the loop connecting α-helices 4 and 5, is suggestive of Sak80α's mechanism to load ssDNA onto its ssDBG (see discussion).

Despite the modest resolution of this region of the map, it could be perceived that Stl2 employs its two loops in the CTD to interact and fix one Sak80α CTD (Figs. 2D and 5F). To explore the significance of these loops, we replaced residues Glu89, Val90, Thr91 in loop 1 (Stl2-Loop 1) and residues Phe195, Pro196, Asn197, Lys198, Gly199, Glu200 in loop 2 (Stl2-Loop 2) by alanine residues. After purifying these new Stl2 versions, we assessed their oligomeric properties in solution by mass photometry (Supplementary Fig. 1C). Both Stl2-Loop1 and Stl2-Loop 2 exhibited two peaks albeit the large peaks corresponded to a slightly smaller mass than the wild-type (Supplementary Fig. 1C). We

then measured the ability of Stl2-Loop 1 and Stl2-Loop2 to bind to their cognate DNA in vitro (Supplementary Fig. 6A). In both cases, these loop mutants exhibited slightly enhanced DNA binding affinity compared to the wild type. We repeated the EMSAs with these mutants but this time including Sak80α and found that both Stl2-Loop 1 and Stl2-Loop 2 remained bound to the DNA in the presence of Sak80α (Supplementary Fig. 6B). Subsequent experiments, including mass photometry and annealing assays, were performed to confirm the role of these loops in stabilizing the Sak80α-Stl2 complex and inhibiting Sak80α. Mass photometry analysis of Stl2-Loop 1 mixed with Sak80α revealed the formation of larger peaks corresponding to 1 and 1.5 MDa similar to the wild-type, while mixing Stl2-Loop 2 with Sak80α did not generate such peaks, thus indicating that the residues mutated in Stl2-Loop 2 were critical for complex formation (Supplementary Fig. 6C). We then repeated the annealing assay. Stl2-Loop 1 showed slightly better inhibition of Sak80α-mediated annealing at low concentrations but slightly worse inhibition at high concentrations compared to the wild-type, whereas Stl2-Loop 2 did not inhibit the annealing reaction in vitro (Supplementary Fig. 6D).

The data regarding the Stl2-Loop 1 mutant seemed puzzling. Despite the presence of Sak80α in the EMSA's reaction mix, the

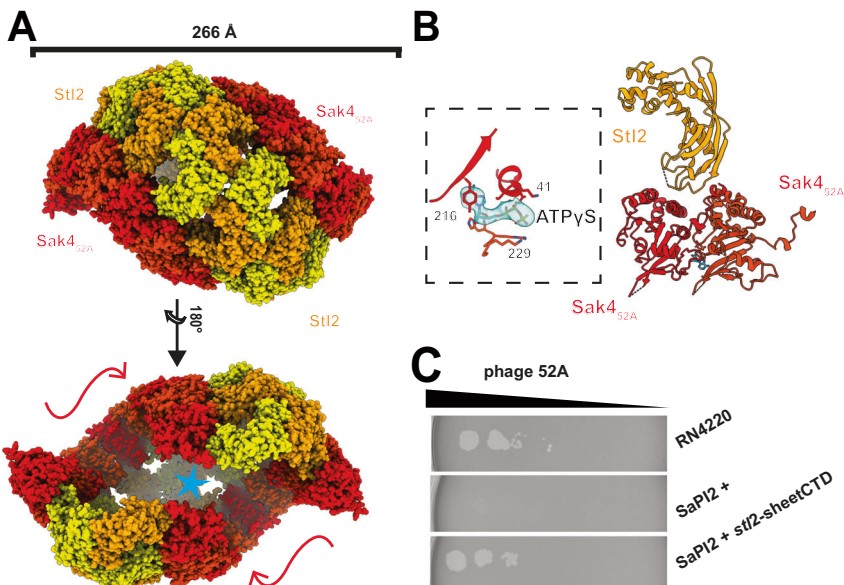

**Fig. 6 | The structure of Stl2·Sak4_{52A}·ATPγS complex. A** Model displayed as spheres, where Stl2 protomers are depicted in yellow and orange whereas Sak4_{52A} filaments in red. The wiggly arrows indicate the opposite disposition of the Sak4_{52A} filaments, and the blue star marks the region where the two filaments collide to each other. **B** Cartoon representation of an Stl2 protomer (chain ID 3F) and two Sak4_{52A} protomers (chains ID 4E and 4F). The nucleotide ATPγS is boxed and zoomed in on the left-hand side, where it is displayed as sticks fitted into the density shown in blue. Side chains of the residues Lys41, Thr42, and Tyr216 in protomer 4E and Lys229, Arg231, and His232 in protomer 4F are shown as sticks. See Supplementary Fig. 7 for chain ID identifiers. **C** Spot assay as in Supplementary Fig. 6F where the strains RN4220, RN4220 SaPI2, and RN4220 carrying the SaPI2 stl2-sheetCTD variant were challenged with the phage 52A. Source data are provided as a Source Data file.

Stl2-Loop 1 mutant remained bound to its cognate DNA. However, the mass photometry and annealing assays indicated that the Stl2-Loop 1 mutant was able to complex with and inhibit Sak_{80α}. Because the conditions between these experiments were different, we reasoned that the in vitro conditions in each experiment could be favoring the opposing observed results. We therefore decided to analyze the ability of Sak_{80α} to release (induce) Stl2-Loop 1 and Stl2-Loop 2 variants from the DNA using the in vivo transcriptional fusion assay, which represents a proper physiological scenario. stl2-loop 1 was partially induced after the over-expression of sak_{80α} with cadmium, whereas stl2-loop 2 showed no induction (Supplementary Fig. 6E). Altogether our structural, biochemical and in vivo data elucidated that (i) Stl2 scaffolds Sak_{80α} rings by binding Sak_{80α}'s CTDs and prevents Sak_{80α} for coordinating with its natural partner SSB_{80α}; (ii) the residues mutated in Stl2 loops 1 and 2 play an important role in modulating the correct affinity for DNA; (iii) while those residues in loop 2 play a fundamental role in binding and inhibiting Sak_{80α}. Importantly, our cryo-EM analysis of the Stl2-Sak_{80α} complex has allowed us to visualize the CTD of a Rad52-like annealase for the first time, revealing different conformations and suggesting a mechanism of action for loading the ssDNA into the annealase's ssDBG (see discussion).

### Stl2 oligomerization as multimers is necessary for its inhibition activity against Sak_{80α}

Our cryo-EM structure of the Stl2-Sak_{80α} complex strongly suggests that the ability of Stl2 to self-oligomerize into circles/filaments is coupled with its inhibitory activity against its target. To confirm this hypothesis, we ran a new set of annealing assays with the previous Stl2 mutants we generated: Stl2-helixNTD and Stl2-sheetCTD. It became evident that the Stl2's inhibition activity was abolished with these two mutants (Supplementary Fig. 6D) demonstrating that the in vitro inhibition mechanism against Sak_{80α} is oligomerization-dependent. Moreover, the stl2 sheetCTD mutant was uninduced by Sak_{80α} as per the transcriptional fusion assays (Supplementary Fig. 6E), suggesting that the ability to oligomerize has emerged as a necessary condition

for the induction mechanism of SaPI2, which is a unique feature among SaPIs. To fully demonstrate the hypothesis that the multimerization of Stl2 as a collar of dimers is linked to its antiviral function, we introduced the stl2-sheetCTD mutation in the SaPI2 genome by standard procedures of allelic replacement[27] and repeated the spot assays in a complete physiological scenario. As shown in (Supplementary Fig. 6F) the stl2-sheetCTD variant was sensitive to phage infection, thus supporting the hypothesis that the multimerization of Stl2 is needed to induce the parasite and mediate the immunity in vivo.

In our view, the fact that Stl2 oligomerizes as rings mimicking Sak_{80α}'s oligomeric state is an unprecedented evolutionary adaptation. The Stl2 apo structure therefore suggests that its ability to oligomerize into large multimeric structures is the basis for its promiscuity in targeting phage-encoded HRs from structurally divergent superfamilies. To demonstrate this, we then turned to Sak4_{52A} the filament-forming Rad51-like recombinase carried by the phage 52A.

### The cryo-EM structure of the Stl2-Sak4_{52A} complex

The Stl2-Sak4_{52A} complex also included ATPγS and the micrographs showed highly homogeneous distribution of particles (Supplementary Fig. 7A). The closest structural homolog of Sak4_{52A} as per Dali server is human Rad51 (PDBID 7C9A, Supplementary Fig. 7F). Due to the symmetry of the Stl2 dimers, the two Sak4_{52A} filaments are disposed in opposite orientations (Fig. 6A). Notably, Stl2 maintains its multimeric disposition, but this time adopts an open washer-like state, whose openness is not sufficiently flexible to decorate two right-handed Sak4_{52A} filaments continuously, and after the completion of one turn the two Sak4_{52A} filaments collide to one another (Fig. 6A). Stl2 introduces the end of the wings (the loops 1 and 2) between Sak4_{52A} protomers so that one Stl2 protomer interacts with two Sak4_{52A} (Fig. 6B). Surprisingly, Stl2 does not interact with nor seem to affect the nucleotide. Once more, the inhibition of Sak4_{52A} by Stl2 appears to hinge on its capacity to oligomerize. To validate this, we conducted another spot assay using the SaPI2 stl2-sheetCTD mutational variant

challenged by phage 52A. Notably, the antiviral immunity against phage 52A was nullified (Fig. 6C).

## Stl2 homologs intra and extra genus

In silico analysis of Stl2 homologs revealed similar transcriptional repressors in species beyond *S. aureus* (Supplementary Fig. 8). Additionally, distant homologs were also found in *Streptococcus*, *Listeria*, *Ureibacillus*, *Mammaliicoccus*, *Fundicoccus*, *Lysinibacillus* and *Klebsiella*, suggesting SaPI2-like elements targeting homologous recombinases are widely spread in nature. Employing AlphaFold2 Multimer algorithm[28,29] unveiled that these homologs fold similarly to Stl2 dimers. Supplementary Fig. 8). Given the high prevalence of phage-parasites in the biosphere[4], it seems reasonable to presume that Stl2 homologs will be eventually found spanning bacteria and archaea domains.

## Discussion

In this study, we have shown that SaPIs, the prototypes of a larger family of phage parasites, may have been underestimated in their ability to provide antiviral immunity. While they might carry anti-phage defense systems[3,30], we demonstrate that they provide robust immunity mediated by their conserved transcriptional repressor, Stl. Using the nowadays standard spot assay, the SaPI-mediated antiviral immunity can be compared with other systems. Notably, SaPI2-mediated immunity resulted in a $10^5$-fold reduction in the spot assays, surpassing other well-established anti-phage defense systems[3,31,32]. In our view, this suggests that the phage-parasite itself is used by the host as an anti-phage system, similar to other phage parasitizing mobile elements such as the phage-inducible chromosomal-like elements, PLEs[33]. This feature may contribute to explain the widespread presence of phage parasites in virtually any microbiome on earth[4].

We have characterized the Stl2-mediated inhibition mechanism targeting phage HRs. The Stl2-Sak$_{80\alpha}$ structure shows for the first time a Rad52-like annealase CTD, which is stabilized by Stl2 in two different conformations, up and down (Fig. 5E). In a scenario where Stl2 is absent, this upward and downward movement suggests how Sak$_{80\alpha}$ might modulate the positioning of its CTD in relation to the ssDBG. It is reasonable to speculate that this motion allows Sak$_{80\alpha}$ to move its CTD away from the ssDBG to interact with the acidic tail of ssDNA-wrapped SSB$_{80\alpha}$. The Sak$_{80\alpha}$ CTD then displaces SSB$_{80\alpha}$ from the ssDNA and the opposite downward movement brings closer or loads the free ssDNA into Sak$_{80\alpha}$'s ssDBG. Alternatively, the movement to bring closer the ssDNA to the ssDBG might precede the displacement of SSB from the ssDNA. The correct sequence of events needs to be elucidated. When Stl2 is present, two Sak$_{80\alpha}$ rings are ensconced within a Stl2 armor, preventing Sak$_{80\alpha}$'s CTD from engaging with anything beyond the Stl2 cage. Moreover, the ssDBG of the Sak$_{80\alpha}$ ring (blue patch in Supplementary Fig. 3A) is likewise entrapped within the Stl2 scaffold becoming invisible (Fig. 5D). The two Sak$_{80\alpha}$ rings are kept separated so that they cannot interact to each other, a proposed intermediate step for efficient complementary ssDNA annealing[34]. The structure of Stl2 apo evidenced its tendency to form ring-shaped and filamentous structures, thus mimicking its target quaternary structure (Fig. 2C). In our view, this represents a remarkable evolutionary structural-functional adaptation. We suspect, however, that this is a non-physiological scenario unless cellular Stl2 is produced in excess during SaPI2 quiescent state, a situation in which rings or filaments would be already waiting for the target to appear. The limitation of the study in this regard prompts us to address this question in future research. Nonetheless, we have demonstrated that key residues at the dimer-dimer interface are necessary for the induction of the SaPI2 island in vivo as evidenced by the transcriptional fusion experiments (Supplementary 6E). Similarly, those key residues mediating dimer-dimer multimerization are also necessary for the inhibition activity against Sak$_{80\alpha}$ in vitro and in vivo, (Supplementary Fig. 6D, F) implying that the

target binding domain comprised by loops 1 and 2 is not sufficient and that the ability to oligomerize as multimers of dimers have emerged as a necessary condition to efficiently provide antiviral immunity against phages carrying Rad52-like annealases. Our Stl2-Sak4$_{52A}$ complex suggests that Stl2 prevents the Sak4$_{52A}$ recombinase from forming long filamentous structures. We have demonstrated that the multimerization nature of Stl2 in a collar of dimers is also coupled with its antiviral immunity against phages carrying Rad51-like recombinases in vivo (Fig. 6C). Hence, this feature is the basis for Stl2 promiscuity against the phage HRs folding in such divergent architectures. The mode of action of Stl2 is distinct from that of the two other characterized Stls (Stl-bov1 and Stl1) and from distant repressors found in other mobile genetic elements. Dimeric Stl-bov1, for instance, targets phage dUTPases that fold as dimers or trimers and dissociate into monomers upon target binding[35]. In contrast, the tetrameric form of Stl1 appears to undergo a conformational change in its DNA binding domain, rendering it incompatible with operator binding[36]. To the best of our knowledge, Stl2's ability to oligomerize into large multimers represents a novel mode of action previously unobserved. In our view, this represents a remarkable example of evolutionary structural-functional adaptation.

In summary, our characterization of Stl2-mediated inhibition marks a new chapter in the study of homologous recombination. Stl2's inhibition mode against homologous recombinases is unique compared to other inhibitors[37–47]. Thus, the characterization and design of Stl2-like inhibitors might bring new biomedical tools in the future. From an evolutionary perspective, SaPI-phage interactions represent a fascinating form of molecular parasitism within bacterial intracellular systems (Fig. 7). Within this type of parasitism, the SaPI's interference with the helper phage must be finely tuned, as the fitness of the helper phage is critical for the successful dissemination of the SaPI. Although the three prototypical SaPIs studied here do not carry canonical anti-phage systems, they do employ various interference mechanisms to disrupt helper phage propagation, showcasing evolutionary strategies beyond classical immunity mechanisms[19–21,48]. In the case of SaPI2, deletion of all phage interference mechanisms it encodes against phage 80α, *ppi*, and *cpm*AB, still significantly impair the reproduction of the phage[20]. We propose that this impairment is mediated by Stl2. Our findings showed that when Stl2 targeted and bound to a plasmid-expressed Sak$_{80\alpha}$ annealase, thus mimicking a *stl2* mutant, followed by a phage 80α infection, immunity was lost in a manner similar to SaPI1Δ*stl* and SaPIbov Δ*stl*. Strikingly, although *ppi* and presumably *cpm*AB were expressed in these *stls* defective variants, they were insufficient to confer phage immunity, unlike when they were expressed from the Pcad exogenous promoter[20]. This contradiction between eutopic *vs* ectopic expression suggests that SaPIs must strike a delicate balance in parasitizing the helper phage under physiological conditions, neither too lenient nor too severe.

Given the conservation of *stls* in these parasites, our work supports the idea that phage-parasites were naturally selected as bacterial defense mechanisms against phages, which contributes to explaining why phage parasites are highly ubiquitous across microbiomes.

## Methods

### Plasmid constructs, cell cultures, protein purification, and allelic replacement mutagenesis

Oligonucleotides, plasmid constructs, and strains used in this study are summarized in Supplementary Tables 2–4 respectively.

*stl2* and the SaPI2-IGR were PCR-amplified from SaPI2-bearing *S. aureus* strain RN4220 genomic material (strain JP2878 provided by the Penadés lab) while *sak$_{80\alpha}$* and *ssb$_{80\alpha}$* were PCR amplified from RN10359 and *sak4$_{52A}$* was PCR amplified from RN4220 lysogenic for phage 52A. *stl2* was cloned into pET-21a between NdeI and XhoI restriction sites, adding LEHHHHHH as extra residues in the Stl2 C-terminus. *sak$_{80\alpha}$* was cloned into a pET-His_1a vector with an

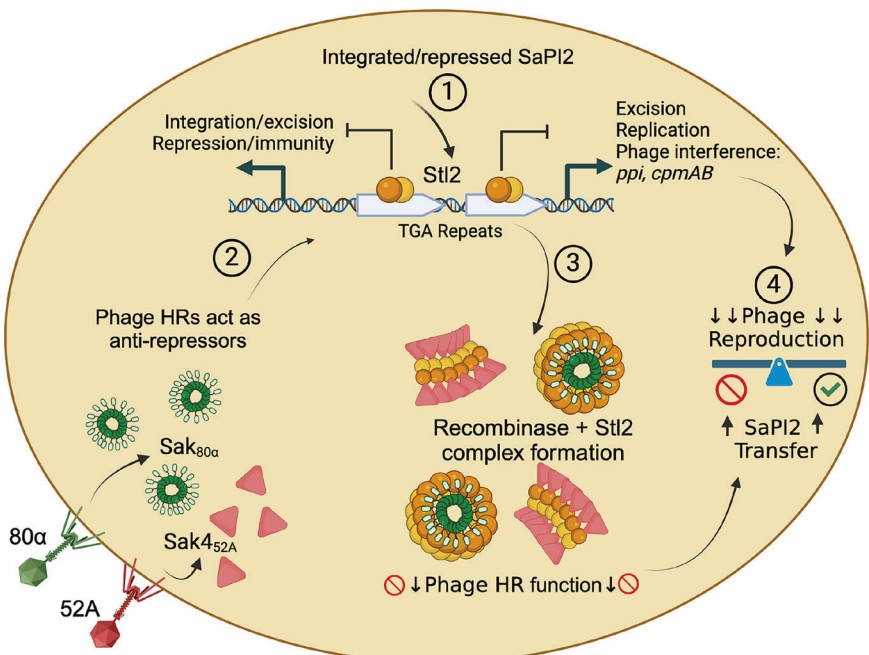

**Fig. 7 | Schematic model of Stl2-mediated immunity against phage infection.** (1) SaPI2 is maintained in a quiescent state by the expression of Stl2, which represses the excision, replication, and phage interference machinery of SaPI2. (2) Helper phages of SaPI2, such as 80α and 52 A, express HRs that inadvertently serve as binding partners for Stl2, triggering SaPI2 mobilization by displacing Stl2 from the operators. (3) Stl2 then forms a stable complex with these HR binding partners, disrupting the function of the phage HRs. (4) The Stl2-mediated inhibition of phage HRs, together with other non-canonical phage interference mechanisms encoded by SaPI2, imposes a balanced impairment on the phage, ensuring the successful spread of SaPI2 and its evolutionary prevalence. Created in BioRender. Mir Sanchis, I. (2025) https://BioRender.com/l51s687.

N-terminal polyhistidine tag followed by a TEV site, between the NcoI and BamHI restriction sites, replacing the GFP-encoding gene normally present in the plasmid. The N-terminal tag for Sak80α consists of residues MKHHHHHHPMSDYDIPTTENLYFQGAMG. $ssb_{80\alpha}$ was cloned into pET-28a between the NdeI and BamHI restriction sites, thus yielding a protein with a N-terminal polyhistidine tag followed by a thrombin site, with residues MGSSHHHHHHSSGLVPRGSH as the N-terminal appendage. $sak4_{52A}$ was cloned into pET-28a between NdeI and XhoI sites, also adding the same appendage to the protein's N-terminus. To solve the Sak80α-Stl2 structure a fourth construct was generated with pETDuet-1, carrying both $stl2$ (NcoI/BamHI sites, reverse oligo encoding for a C-terminal polyhistidine tag) and $sak_{80\alpha}$ (BglII/XhoI sites, untagged). All baseline clones were transformed to DH5α $E. coli$. All mutants were generated by site-directed mutagenesis PCR, followed by digestion with DpnI and transformation to DH5α. All constructs were finally transformed to BL21-(DE).

For large-scale protein expression, 1.5-3 L cultures were grown in 2 L GL-45 bottles with a LEX-48 Bioreactor system (Epiphyte3). The cultures were grown in LB medium supplemented with 100 μg/ml of ampicillin (for strains carrying pET_21a or pET_Duet) or 50 μg/ml kanamycin (for strains carrying pET_28a or pET_His-1a), along with the addition of Antifoam 204 (SKU: A8311-50ML, Sigma-Aldrich). All cultures were grown at 37 °C until the $OD_{600}$ reached a range of 0.6–0.8. At this point, the temperature was lowered to 18 °C, 0.5 mM IPTG was added to induce expression, and the cultures were left to grow overnight. Subsequently, cultures were harvested by centrifugation and the pellets were stored at −80 °C until needed.

Prior to purification, pellets were resuspended in buffer A1 (50 mM sodium phosphate, 5% glycerol, 1 mM DTT, 1 M NaCl, pH 7.0) with a protease inhibitor cocktail (Pierce Protease Inhibitor Mini Tablets, EDTA Free, Thermo Scientific). The suspension was then lysed with 200 μg/ml lysozyme at 37 °C for 30 min, followed by sonication and centrifugation. The supernatant was filtered and loaded onto a HisTrap HP (Cytiva) column pre-equilibrated with buffer A1 with 25 mM imidazole. The proteins were then eluted with a gradient of 25-500 mM imidazole. Peak fractions were checked by SDS-PAGE for the presence of proteins of interest, pooled and dialyzed in either buffer Stl-A2 (0.5 mM EDTA, 100 mM NaCl, 5% glycerol, 1 mM DTT and 20 mM Bis-Tris pH 6.2; for Stl2), or regular buffer A2 (0.5 mM EDTA, 100 mM NaCl, 5% glycerol, 1 mM DTT and 20 mM Tris-HCl pH 7.0) for $SSB_{80\alpha}$ and Sak80α and Sak4$_{52A}$. The proteins were then loaded onto a HiTrap Heparin HP column (Cytiva) equilibrated with the appropriate A2 buffer and eluted with a NaCl gradient up to 1 M.

Peak fractions were again confirmed by SDS-PAGE, pooled, concentrated, and injected into a Superose 6 Increase 10/300 GL column equilibrated with buffer Stl-A3 (20 mM Tris pH 8.0, 5% glycerol, 0.5 mM EDTA, 200 mM NaCl, 2 mM DTT; for Stl2) or regular buffer A3 (20 mM Tris pH 7.0, 100 mM NaCl, 2 mM DTT) for $SSB_{80\alpha}$, Sak80α and Sak4$_{52A}$. Peak fractions were once more confirmed by SDS-PAGE, pooled and concentrated. Stl2 and $SSB_{80\alpha}$ proteins were further supplemented with glycerol to 20%, then all proteins were aliquoted and flash-frozen in liquid nitrogen before being stored at −80 °C. The purified WT proteins and its variants are shown in Supplementary Fig. 9.

The pMAD construct to generate the SaPI2 $stl2$-sheetCTD mutant was generated as follows. A 1kbp section encoding for 500 bp up and down of the sheetCTD mutation site in $stl2$ was PCR-amplified from SaPI2 genomic DNA and cloned between the BamHI and EcoRI restriction sites in pMAD. The ligated vector was transformed into DH5α $E. coli$, purified, and then used as a template for site-directed mutagenesis PCR with the same primers as the ones used to create the sheetCTD mutant in pET21a-$stl2$. The PCR product was DpnI-digested and again transformed to DH5α, from where the resulting plasmid was purified, electroporated to SaPI2-carrying RN4220 $S. aureus$, and used for in vivo allelic replacement. To confirm the presence of the sheetCTD mutation, genomic DNA from the mutated strain was isolated and used as a template for a PCR with primers annealing outside of the pMAD-inserted region. The PCR product was then Sanger-sequenced and digested with MunI, which is a restriction site absent

within wild-type *stl*2 but which is encoded in the sheetCTD mutant region.

## Spot assays

Overnight cultures of the strain RN4220 carrying none or SaPI1, SaPI1Δ*stl*, SaPI2, SaPIbov1, SaPIbov1Δ*stl* were diluted to $OD_{600} = 0.3$, mixed with 3 ml of phage top agar and poured on top of phage base plates. Then 4 µl of the tenfold serial dilutions from different phage lysates were deposited on top and the plates were incubated at 37 °C for 16 h. The same procedure was followed for SaPI2 *stl*2-sheetCTD variant.

For the spot assays in Fig. 1B, bottom part, the overnight cultures of the strains RN4220, RN4220 harboring SaPI2 (RN4220 SaPI2 +), RN4220 SaPI2+ porting empty pGMT3 (Pcad-empty) and RN4220 SaPI2+ carrying pGMT3-FLAG-*sak*$_{80α}$ (Pcad-*sak*$_{80α}$) were diluted to $OD_{600} = 0.1$ and induced with 5 µM $CdCl_2$ at 37 °C until the OD reached 0.2, at which time 150 µL of each culture were mixed with 3 mL of phage top agar and poured on phage base plates. Each plate was then spotted with 4 µL of ten-fold serial dilutions of phage 80α lysates and incubated overnight at 37 °C.

## Electrophoretic mobility shift assay (EMSA)

The proteins were initially diluted in dilution buffer (20 mM Tris-HCl pH 7.0, 5% glycerol, 100 mM NaCl, 0.5 mM EDTA, and 2 mM DTT) and then incubated for 30 min at room temperature in a solution containing 2 nM of FAM-labeled DNA, 20 mM Tris-HCl pH 7.0, 5% glycerol, 100 mM NaCl, 50 ng/µl BSA, 1 ng/µl salmon sperm DNA and 5 mM DTT. EMSAs with Stl2 were supplemented with 5 mM $CaCl_2$, while Sak$_{80α}$ reactions contained 2 mM $MgCl_2$ and those with SSB$_{80α}$ did not contain metal co-factors. For Stl2, the full SaPI2-IGR region was used as a 212 bp PCR product amplified from strain JP2878 with oligos 1_SaPI2-Stl_FP_full-F-FAM and 2_SaPI2-Stl_FP_full-R. As a negative control, a 318 bp region corresponding to the SaPI5 origin of replication was PCR-amplified using oligos FAM-Ori_SaPI5_F and Ori_SaPI5_HindIII_R. For EMSAs with SSB$_{80α}$ and Sak$_{80α}$ a FAM-labeled 48nt ssDNA oligo was used (FAM-oligo-26) and the same oligonucleotide annealed with its complement (oligo-25) as dsDNA. For EMSAs where a gradient of Sak$_{80α}$ was present, 2 mM $MgCl_2$ and 5 mM $CaCl_2$ were included in the reaction and 50 nM (Fig. 5 H) and 100 nM (Fig. 4 A) of the Stl2 variants were incubated with 2 nM FAM-labeled PCR of the SaPI2-IGR, followed by the addition of a Sak$_{80α}$ gradient 0, 50,100, 200, 400, 600, and 1000 nM, excluding 1000 nM concentration in Fig. 5H. All EMSA reactions were incubated at room temperature for 30 min. After incubation, 6x loading buffer (20 mM Tris-HCl pH 8.0, 30% glycerol, 30 mM EDTA, and 0.03% Xylene Cyanol/Bromophenol Blue) was added to the reaction mix and 10 µl were loaded onto a polyacrylamide gel (4% for Stl2 and Sak$_{80α}$, 8% for SSB$_{80α}$) and run in 0.5x TBE at 100 v for 60–90 min (4 °C for Stl2, room temperature for SSB$_{80α}$ and Sak$_{80α}$). The results were visualized with an Amersham Typhoon laser scanner (Cytiva) with the cy2 emission filter selected. Quantification of the bands of each gel was performed with ImageJ and calculation of the $K_d$ and the Hill coefficient ($n$) was analyzed by fitting with the Hill equation using GraphPad Prism 9.3.1, where applicable.

## Fluorescent DNAse I footprinting assay

The same 212 bp PCR product encoding for the SaPI2-IGR as in the Stl2 EMSAs was employed, with the exception that two DNA products were used: one which was 5′ FAM-labeled in the forward oligo and a second where the reverse primer was 5′ FAM-labeled instead. Stl2 was incubated with 600 fmol of IGR-DNA at 37 °C for 30 min in a reaction containing 2 mM $MgCl_2$, 25 ng/µl BSA, 20 mM Tris-HCl pH 7.0, 100 mM NaCl, 5% glycerol and 5 mM DTT. After incubation, 20 µl of sample was mixed with 3 µl of DNAse I (diluted 10-fold: 0.3 µl DNAse I + 0.3 µl 100 mM $CaCl_2$ + 2.4 µl DNAse I reaction buffer, New England Biolabs) at room temperature for either 60 s (naked DNA) or 90 s (DNA + Stl2).

The DNAse reaction was then stopped by adding 200 µl dissociation buffer (0.5 M ammonium acetate, 10 mM magnesium acetate, 1 mM EDTA, 0.1% SDS). 5 µl of 10 mg/ml salmon sperm DNA as well as 0.3 µl glycogen per ml were added to the dissociation buffer prior to use. The reactions were phenol-chloroform purified and resuspended in 10 µl blue formamide loading buffer (95% formamide (v/v), 10 mM EDTA pH 8.0, 0.1% (w/v) Xylene Cyanol/Bromophenol Blue). Prior to loading, both the samples and the G + A sequencing ladder were heated to 95 °C for 5 min. 5 µl of G + A ladder and each sample was loaded and run in a 5% sequencing gel (5% acrylamide, 7 M urea, 25% formamide) which had been pre-run for 1 h. The result was recorded with an Amersham Typhoon laser scanner (Cytiva) with the cy2 emission filter. The G + A sequencing ladder used in the footprinting experiments was produced as follows: 8 µl of the same FAM-labeled DNA used in the footprinting assay was mixed with 2 µl of 1 mg/ml salmon sperm DNA and then incubated at 37 °C for 25 min after addition of 1 µl 4% formic acid (Sigma-Aldrich). 150 µl of 1 M piperidine (Sigma-Aldrich) was added to the mix and then incubated at 90 °C for 30 min. The ladder was then dried at 95 °C, after which 40 µl of deionized water was added, and then dried again. Finally, the ladder was resuspended in blue formamide loading buffer to the same concentration as the DNA in the footprinting samples.

## Nitrocefin-based β-lactamase in vivo assay

For the nitrocefin assay in Fig. 1B, the section of SaPIs 1, 2, and bov1 encompassing *stl* to *ppi* were cloned between restriction sites BamHI and KpnI in pCN41, transcriptionally fusing *blaZ* to the region downstream of *ppi*. For SaPI1, the origin of replication (*ori*1) was left out of the insert (via overlap PCR, see supplemental tables for the description) due to plasmid stability issues in *E. coli*.

For the nitrocefin assay in Supplementary Fig. 1E, a section of SaPI2 encompassing *stl2*, the SaPI2 intergenic region, *str2*, and half *xis* gene was cloned into pCN41 between restriction sites BamHI and KpnI, transcriptionally fusing the vector-encoded *blaZ* to the *xis* gene, generating pIMS1473 (pCN41$_{stl2-str2}$). Further *stl*2 mutations were generated by site-directed mutagenesis PCR using pIMS1473 as a template and then transformed into DH5α *E. coli*.

All pCN41 derivates were subsequently electroporated into RN4220 *S. aureus*.

For the nitrocefin assay in Fig. 4B, the same N-terminally FLAG-tagged *sak*$_{80α}$ construct used for the spot assays in Fig. 1B was employed here. RN4220 electroporated with this construct was used as recipient to co-transform all the pCN41$_{stl2-str}$ variants. Strains carrying both plasmids were selective for erythromycin (10 µg/ml, from pCN41) and chloramphenicol (10 µg/ml, from pGTM3). For the β-lactamase assays, overnight cultures were diluted in TSB without antibiotics to a starting $OD_{600}$ of 0.1 and then grown in a shaker incubator at 37 °C, 220 RPM, until they reached an $OD_{600}$ of 0.2. At this point, expression of FLAG-tagged Sak$_{80α}$ in pGTM3 was induced with 5 µM $CdCl_2$, and all cultures grown for a further 2 h. After induction, 50 µl of each culture was loaded onto a 96-well plate and β-lactamase activity was measured by addition of 50 µl 137 µg/ml nitrocefin and then recording absorbance values at 490 nm and 600 nm every 45 s for either 30 min (*stl2-str*2 variants) or 1 h (*stl-ppi* variants) with a Multiskan SkyHigh Microplate Spectrophotometer (ThermoFisher). All data points were normalized by the initial $Abs_{600nm}$ value. β-lactamase activity units are defined as $[(\Delta slope*100)/t]$ where $\Delta slope$ is obtained by subtracting the $Abs_{490/600nm}$ value from the end of the slope by the initial $Abs_{490/600nm}$ value and $t$ is the time duration of the slope in minutes.

## Annealing assays

2 nM of oligo-25 (Supplementary table 2) was pre-incubated for 10 min at room temperature with 200 nM tetrameric SSB$_{80α}$ in a reaction mix containing 20 mM Tris pH 7.0, 100 mM NaCl, 5% glycerol, 5 ng/ml BSA

and 1 mM DTT. At the same time, 50 nM $Sak_{80\alpha}$ (17-mer concentration i.e., 850 nM monomeric) and an Stl2 gradient were also mixed and pre-incubated at room temperature for 10 min. After pre-incubation, the $Sak_{80\alpha}$-Stl2 mix was added to the $SSB_{80\alpha}$-oligo-25 mix, with the reaction being started upon addition of 2 mM $MgCl_2$ and 2 nM FAM-labeled oligo-26 (complementary to oligo-25). After 5 min the reaction was stopped by adding 4 µl of 6x stop/load buffer containing 6 µM unlabeled oligo-26, 1.5% SDS, 120 mM Tris-HCl pH 7.0, 30% glycerol, 90 mM EDTA pH 8.0 and 0.03% Xylene Cyanol/Bromophenol Blue. For reactions with only $Sak_{80\alpha}$, oligo-25 was pre-incubated with the indicated $Sak_{80\alpha}$ concentration as 17-mer together with 2 mM $MgCl_2$ in the reaction mix for 10 min at room temperature. FAM-oligo-26 was then added, and the reaction stopped after 5 min. The reactions were then run in 8% poly-acrylamide gels (19:1 acrylamide/bis-acrylamide) and the resulting bands visualized with an Amersham Typhoon laser scanner (Cytiva), again with the cy2 emission filter. For each lane, the areas under the curve of ss- and dsDNA bands were quantified with ImageJ, and the dsDNA percentage was obtained by dividing the dsDNA area by the total (ssDNA + dsDNA) area values. Triplicate values were then input into GraphPad Prism 9.3.1, which was used to generate the resulting graphs. For the permutated reactions between 200 nM of $SSB_{80\alpha}$ (tetrameric), 50 nM $Sak_{80\alpha}$ (heptadecameric), and 500 nM Stl2 (dimeric). Oligo-25 was pre-incubated with SSB in reactions where it was present as well as $Sak_{80\alpha}$ and Stl2, which were pre-incubated together when both were present.

## Mass photometer (MP)

$Sak_{80\alpha}$ and Stl2 (or indicated variants) were mixed at a 1:1 molar ratio and then incubated at 37 °C for 30 min. Prior to measurement, the protein solutions were diluted in dilution buffer (20 mM Tris pH 8.0, 0.5 mM EDTA, 200 mM NaCl, 2 mM DTT) to a working concentration between 50 to 100 nM, depending on the dissociation properties of each individual sample. Microscope coverslips (No. 1.5H, 24 × 50 mm) were sequentially rinsed with isopropanol and Milli-Q water three times and dried with a clean compressed-air stream. CultureWell™ gaskets (GBL103250-10EA, Sigma-Aldrich) were cut to cover an area of four wells (3 mm diameter, 1 mm deep) and placed in the center of the clean coverslip, ensuring a tight fit through gentle pressure. A droplet of immersion oil (Carl Zeiss™ Immersol™518 F, Fisher Scientific) was applied to the objective of the flow-chamber of One$^{MP}$ Mass Photometer (Refeyn Ltd, Oxford, UK), and the prepared coverslip-gasket assembly was mounted and stabilized with small magnets. Each protein was measured in a new well. To find the focus, fresh dilution buffer was loaded in wells of the gasket, and focus was determined by the autofocus system. 15 µl of protein sample was loaded onto the well for the measurement. Data acquisition was performed using AcquireMP (Refeyn Ltd, v 2.4.1), and movies lasting 60 s were recorded. All MP movies were processed and analyzed using DiscoverMP (Refeyn Ltd, v 2.4.2). Bovine Serum Albumin was used as the standard sample.

## Specimen preparation for Cryo-EM

**$Sak_{80\alpha}\Delta CTD$.** Sample preparation of $Sak_{80\alpha}\Delta CTD$ was done by diluting the stock protein to 1.5 mg/ml in solution containing 20 mM Tris-HCl pH 7.6 and 100 mM NaCl and 1 mM DTT. 4 µl of the sample was applied to a glow-discharged QuantiFoil R 2/1 300 mesh copper grid, blotted for 4 s and then plunge-frozen. All samples were plunge-frozen in liquid ethane at 4 °C and 100% humidity with a Vitrobot Mark IV (Thermo Fisher Scientific).

**$Sak_{80\alpha}$-Stl2.** To solve the cryo-EM structure of the Stl2- $Sak_{80\alpha}$ complex, we cloned and concomitantly purified them both from pETDuet-1 expression vector where full-length $Sak_{80\alpha}$ and Stl2 were untagged and C-terminally His-tagged respectively. The protein complex was plunge-frozen immediately after the last step of the co-purification pipeline. In this regard, after passing the $Sak_{80\alpha}$-Stl2 complex through a Superose 6 Increase 10/300 GL column (Cytiva) the concentration of the fraction corresponding to the third elution peak was measured at 280 nm with a NanoDrop set to 1 Abs = 1 mg/ml and then diluted it to 1.5 mg/ml. Here, QuantiFoil R 2/1 300 mesh copper grids were used, with the sample being blotted for 2 s before plunge-freezing.

**Stl2 apo.** Stl2 was diluted into 5 mg/ml in buffer containing 20 mM Tris pH 8.0, 0.5 mM EDTA, 0.2 M NaCl, 1.3% glycerol, and 2 mM DTT. 4 µl of Stl2 was then applied on a QuantiFoil R 2/1 300 mesh copper grid coated with a 2 nm carbon layer. The blotting condition was the same as described previously, except the blotting time and blot force were 5 seconds and −5, respectively.

**Stl2-Sak4$_{52A}$-APTγS.** Both Sak4$_{52A}$ and Stl2 were mixed and incubated at a 1:1 molar ratio (with Sak4$_{52A}$ at 1 mg/ml) in buffer containing 20 mM Tris, pH 7.6, 100 mM NaCl, 1 mM DTT and 2 mM ATPγS for 10 min at 37 °C. 4ul of sample was then applied to a QuantiFoil R 2/1 300 mesh copper grid and blotted for 3.5 s before plunge-freezing.

## Cryo-EM data collection

For $Sak_{80\alpha}\Delta CTD$, Stl2 apo, and Stl2-Sak4$_{52A}$-APTγS, untilted datasets were collected using a Glacios Cryo Transmission Electron Microscope. For $Sak_{80\alpha}$-Stl2, two datasets were collected on the same grid with a Titan Krios microscope, untilted and 30° tilted. While both microscopes were equipped with a Falcon4i Direct Electron Detector, Glacios was operated at 200 kV with a 100 µm objective aperture while Titan Krios was operated at 300 kV and equipped with a Thermo Scientific Selectris energy filter. The Glacios and Titan Krios datasets were collected with a magnification and nominal pixel sizes of 190,000x, 0.756 Å/pix, and 165,000 x, 0.704 Å/pix, respectively. All data collection parameters are summarized in Supplementary Table 1.

## Image processing and 3D reconstruction

*$Sak_{80\alpha}\Delta CTD$.* The workflow for $Sak_{80\alpha}\Delta CTD$ is illustrated in Supplementary Fig. 4. Initial data processing was carried out in the CryoSPARC 4.2.1 Live[49] environment, where 4344 raw images were motion-corrected and CTF-estimated on-the-fly during data collection and then filtered based on CTF fit estimation and relative ice thickness. Initial particle selection was carried out in a template-based manner after generating 2D templates from a previous $Sak_{80\alpha}$ wild-type 3D map. Around 1.5 million particles were extracted with a 400px box size and exported to the full CryoSPARC environment, of which 144 173 were selected after three rounds of 2D classification, where only 18-mer top views and side views of similar diameter were selected. From this, an initial model was generated, and one round of 3D classification was carried out. The best classes were selected and subjected to two further rounds of 2D classification, resulting in a stack of 30 688 particles. Using this set for homogenous refinement of the initial model and applying C18 symmetry resulted in a 3.47 Å map. To improve the quality of the map, template picking was again used with the new stack of particles, followed by three rounds of 2D classification, producing a new stack of 126 696 particles. From there the ab initio job was run again, mixing the newest and previous particle sets, followed by another round of 3D classification, yielding in a new set of 87 992 particles. Homogenous refinement with this improved dataset produced a 3.24 Å map at C18 symmetry. The particle set was then subjected to symmetry expansion, multiplying the particle stack by 18 to 1,365,300 particles, then used for local refinement, generating a final map at 3.21 Å which was sharpened with the inverse of the job's indicated Guinier plot B-factor (−136.2) and then used for model-building. For structural modeling of $Sak_{80\alpha}\Delta CTD$, the wild-type protein's AlphaFold2[28] prediction as a monomer was docked in ChimeraX[50,51] with residues D141-Q207 having been deleted. The model was then subjected to one round of real-space refinement in PHENIX[52] and repeatedly manually adjusted in Coot[53] and validated with PHENIX.

***Stl2***. Data processing was performed using CryoSPARC/CryoSPARC Live[49]. In detail, the movie frames were first aligned and superimposed with patch motion correction. Contrast transfer function parameters were calculated by patch CTF estimation. Blob picker was first used to pick the particles in 200 micrographs. The particles were then used for generating several 2D averages which were subsequently used as a template for template picking. Template picking was then performed for the whole dataset, and 1,348,125 particles were picked. Several rounds of 2D classification were performed to remove the bad classes. Ab initio reconstruction was performed to generate 4 classes, and a good class containing 598 680 particles was further processed by 3D classification. 5 classes (594 605 particles) out of 6 classes particles were used in 3D homogenous refinement, generating a 3.91 Å map. D1 symmetry was applied to generate a final map with the resolution of 3.69 Å. The resolution estimation was based on the gold standard Fourier shell correction (FSC = 0.143) criterion. Local resolution was estimated with CryoSPARC. For modeling, the AlphaFold prediction[28] was first used to predict the Stl2 dimer. The map was auto-sharpened by PHENIX[52]. The predicted structure was then fitted into the map, manually corrected with Coot[54], and auto-refined in PHENIX[52].

***Stl2-Sak$_{80\alpha}$ complex***. For Stl2 -Sak$_{80\alpha}$ (workflow illustrated in Supplementary Fig. 5) two datasets were collected; an initial one of 16 539 micrographs and a second one, tilted at 30°, consisting of 11,033 micrographs. Both datasets were processed separately in CryoSPARC 4.2.1. First the micrographs were imported, then motion corrected with Patch Motion Correction, and CTF-estimated with Patch CTF Estimation. The processed micrographs were filtered by CTF Fit resolution (<5 Å). A particle set resolving to a 6 Å map obtained from a prior Sak-Stl dataset was used as an initial template for template picking (particle diameter set to 400 Å). From the untilted dataset, 1,157,383 particles were obtained after extraction (700px, binned to 200px). For the tilted dataset, 733,641 particles were obtained once extracted (also 700px binned to 200px). Each particle set was then subjected to two rounds of 2D classification, selecting only classes where two Sak$_{80\alpha}$ rings of similar intermediary diameters could be seen (see Supplementary Fig. 5), resulting in an untilted particle set of 296 168 particles and a tilted one with 215 801 particles. Both sets were then used for separate initial model jobs (set to generate 5 ab initio classes each). The particles yielding the best initial model (120 869 untilted particles, 89 123 tilted particles) from each job were then re-extracted to 700px (unbinned), merged, and homogenously refined at D1 symmetry, generating a map at 4.02 Å. The particles were then 2D-classified, re-extracted to 500px box size, homogenously refined (4.07 Å, D1 symmetry) symmetry-expanded to D1, and used for local refinement, resulting in a final map with a reported resolution of 3.75 Å, which was then sharpened with the inverse of the job's B-factor value (−72.2). The Sak$_{80\alpha}$-Stl2 model was fitted with Coot and Refined in PHENIX. The final model consisted of 26 Sak$_{80\alpha}$ protomers where their CTDs were modeled, 8 Sak$_{80\alpha}$ protomers where their CTDs were left out (totaling 2 heptadecameric rings) and 30 Stl2 protomers (totaling 15 dimers).

***Stl2-Sak4$_{52A}$-ATPγS complex***. For this complex (workflow illustrated in Supplementary Fig. 7), micrographs were again motion-corrected and CTF-estimated on the fly with CryoSPARC Live while selecting for CTF fit (<5 Å) and relative ice thickness (<1.16). The remaining exposures (1239 out of 1301) were then used for blob picking, where particle picks were filtered by power score (between 135 and 1130), resulting in an initial particle stack of 142,629 extracted particles with a box size of 700 px, binned to 200 px. After one round of 2D classification, 111,687 particles were kept, which were used to generate 5 initial models (after unbinned re-extraction). The best volume was used for homogenous refinement with all input particles, yielding a 3.98 Å model at D1 symmetry. The initial blob-picked particle set was then used for another round of 2D

classification, where 118 761 particles were selected and used for template picking (230 Å particle diameter). The resulting particle coordinates were extracted to 700 px and subjected to 2D classification, resulting in a second stack of 150,076 particles. These were then used for generating an additional 5 initial maps. The best map and its accompanying particles (39,234) were homogenously refined to 3.98 Å at C1 symmetry and 3.72 Å at D1 symmetry. Both the 111 687 and 39 234 particle sets were then merged and re-extracted with a box size of 500 px and used for refining the 3.72 Å D1 map (cropped to 500 px) to 3.37 Å. The particle set was then D1 symmetry-expanded, doubling the particle set to 283 726, used for local refinement, resulting in a final D1 map at 3.35 Å which was then sharpened by the inverse Guinier plot B-factor (−68.5).

For model building, the AlphaFold predictions of both Sak4$_{52A}$ and Stl2 were docked into the map with ChimeraX and then real-space refined in Phenix.

## Search and alignment of Stl2 homologs

PBlast homology search was conducted using Stl2 sequence as query. Among the retrieved sequences, twenty proteins whose sequence showed homology beyond Stl2's DNA binding domain were selected for sequence alignment using MUSCLE 3.8.425 via Geneious Prime 2022.2.2 interface. The accession numbers are the following: *Staphylococcus warneri*, WP_251511716; *Staphylococcus caprae*, WP_285118797; *Staphylococcus epidermidis*, WP_203078939; *Staphylococcus simulans*, WP_023015903; *Staphylococcus gallinarum*, WP_232139536; *Staphylococcus equorum*, WP_285323974; *Staphylococcus xylosus*, WP_058625842; Staphylococcus *haemolyticus*, WP_186294988; Staphylococcus hominis, WP_065428529; *Mammaliicoccus sciuri*, WP_231493126; *Mammaliicoccus fleurettii*, WP_203153645; *Staphylococcus pettenkoferi*, WP_088606452; *Staphylococcus agnetis*, WP_252584838; *Fundicoccus ignavus*, WP_153861209; *Lysinibacillus sp*, WP_082337382; *Mammaliicoccus lentus*, WP_301420198; *Klebsiella pneumoniae*, PCQ20268; *Listeria innocua*, EIX7078147; *Streptococcus pyogenes*, WP_136309926; *Ureibacillus galli*, WP_191706115.

All figures and cartoons throughout the manuscript were generated with ChimeraX, BioRender or Genious Prime 2022.2.2 (https://www.geneious.com). Except for CryoSPARC and BioRender, the different software packages used were provided by SBGrid[55].

## Statistical information

Data are presented as individual dots on top of the box-whisker-plot where the whiskers represent the minimum and the maximum values, the borders of the box represent the 25th and the 75th percentile, while the horizontal line crossing the box represents the median. (*) $P$ value < 0.05 was considered statistically significant. Three independent biological replicates ($n = 3$) with each having three experimental replicates were used in either one way ANOVA followed by post hoc test for multiple comparisons or an unpaired two-tailed Student's t-test performed in GraphPad Prism 10.4. N values are referred to biological replicates, whereas technical replicates refer to repeated analysis of the same sample. The statistical analysis is based on averaged values across biological replicates and is not derived from pooled technical and biological replicates.

## Reporting summary

Further information on research design is available in the Nature Portfolio Reporting Summary linked to this article.

## Data availability

Source data are provided with this paper. All maps and models generated in this study have been deposited in the following public databases under the following accession codes: Stl2-apo (EMD-18248 and PDB 8Q86); Sak$_{80\alpha}$ CTD-apo (EMD-17821 and PDB 8PQ8); Sak$_{80\alpha}$-Stl2 (EMD-18346 and PDB 8QE9); Sak4$_{52A}$-Stl2 (EMD-19048 and PDB

8RC5). See Supplementary Table 1 for data collection and refinement parameters. Source data are provided with this paper.

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

## Acknowledgements

We thank the Knut and Alice Wallenberg Foundation, members of the Wallenberg Centre for Molecular Medicine Umeå, Umeå University and Region Västerbotten. We thank Michael Hall and Camilla Holmlund for their help in specimen preparation, screening, and data collection at the Umeå Core Facility for Electron Microscopy, a node of the Cryo-EM Swedish National Facility, funded by the Knut and Alice Wallenberg, Family Erling Persson and Kempe Foundations, SciLifeLab, Stockholm University, and Umeå University. Initial computations were enabled by resources provided by the Swedish National Infrastructure for Computing (SNIC) at HPC2N partially funded by the Swedish Research Council through grant agreement no. 2018-05973. We thank all members of the Medical Biochemistry and Biophysics Department, Anna Åberg for her help with the footprinting assay, and the Hofer's lab for helping with the mass photometry. We thank Thyra Boafo for her hands during her rotation in the lab.

## Author contributions

Conceptualization, I.M-S.; Methodology, G.D-A., C.Q., A.S, N.L., I.M-S.; Investigation, G.D-A., C.Q., A.S., N.L., I.M-S.; Writing—Original Draft, G.D-A., C.Q., I.M-S.; Resources, Supervision and Funding Acquisition, I.M-S.

## Funding

## Competing interests

The authors declare no competing interests.
