## [Transparent Peer Review file · Nature Communications]

Phage parasites targeting phage homologous recombinases provide antiviral immunity

Corresponding Author: Dr Ignacio Mir-Sanchis

Version 0:

Reviewer comments:

Reviewer #1

(Remarks to the Author)

In this study, Debiasi-Anders et al. have characterized to the structural level the interaction between the SaPI2 repressor protein, Stl2 with two different families of recombinases, Rad52-like Sak80 α and Rad51-like Sak452A. The authors have used a wide range of biochemistry techniques, in vivo analysis and they have produced structural data that provide strong evidence of these interactions. Since the discovery of these phage parasites and specially in the last 10 years, much work has been produced to elucidate the molecular interactions between the SaPIs and their helper phages. The work presented in this manuscript adds up to that knowledge by elucidating how SaPI2 interfere with their targeted phage. The detailed structural data provided in this work by solving the structure of the Stl2 repressor, shows it oligomerize as rings/filaments to mimic the folding of their targets: Sak80 α (rings) and Sak452A (filaments). The authors further confirm these interactions by solving the complexed structures of the Stl2 with Sak80 α and Sak452A. The authors also show that these interactions impact the helper phage ability to infect and propose this mechanism as another way SaPIs can provide immunity without encoding canonical immune systems. Given the high interest in the present time about the study of the bacteria immunity systems arsenal, this work could be of potential interest to the microbiology community. Overall, the manuscript is well-written, the rationale for each experiment is well laid out and the results are sound. There are few comments for the authors to be addressed before the manuscript could be accepted for publication:

Abstract, page 1, lines 13-15: "Unrelated anti-phage defense systems target phage-encoded homologous recombinases (HRs). The inhibition mechanisms of anti-phage systems targeting phage encoded HRs remain uncharacterized." Which are these anti-phage defense systems targeting HRs? Other than in the Abstract there is no mention to other anti-phage defense systems targeting HRs through the text.

Results, page 2, lines 30-31: "To investigate this, we utilized variants of SaPI1 and SaPIbov1 in which the stl genes have been inactivated through genomic mutagenesis: SaPI1 Δ stl, SaPIbov1 Δ stl." It is strange to me why the authors following the rationale for this experiment did not include SaPI2 Δ stl in this experiment. The authors hypothesized that the strong interference phenotype for SaPI2 is due to Stl2, but without testing the actual Stl2 mutant to corroborate their hypothesis.

Results, page 2, lines 34-38: "Surprisingly, we discovered that the absence of the stl gene in SaPI1 Δ stl and SaPIbov1 Δ stl variants turned them sensitive to phage infection, indicating that constitutive expression of SaPI-encoded genes in these variants was not sufficient for phage protection. Moreover, this result suggested the necessity of stl for full immunity (Figure 1B)."

This result is indeed surprising, since it is known that these SaPIs although not carrying "canonical" anti-phage defense systems, they do carry different factors to interfere with the phage reproduction such as ppi (interfere with helper phage DNA packaging), cpmA and cpmB (capsid size redirection genes) or ptiA, ptiB and ptiM (interference with helper phage late gene expression). Do the authors have a hypothesis as to why in a mutant in stl, where these factors should be constitutively expressed, they do not provide immunity?

Results, page 4, lines 21-22: "We struggled to obtain a good quality cryo-EM map of full length Sak80a, consequently, we decided to focus on a truncated version of Sak80a where its CTD was removed" It would be important to explain the rationale behind the decision to remove the CTD domain, given that later it is shown as the part that interacts with Stl2.

Results, page 4, line 27: what does SSAPs stands for?

The discussion is quite concise, but in my opinion, it was missing some thoughts about the Gp2.5-like family of helicases that also interacts with Stl2. Do the authors have any hypothesis as to which kind of interactions between Stl2 and this family of helicases could be occurring? Would Stl2 be also mimicking the structure of Gp2.5-like helicases?

Methods, page 18, line 28-29: "All maps and models have been deposited in public databases." Please include in this section the PDB ID for the structures deposited in this manuscript.

Minor comments:

There are some images with poor quality or poor resolution when zoomed in: Figure 1.A (maps' SaPIs with IGR), Figure 4.B (β -gal activity graph) and 4.C (mass photometry table), Extended Data Figure 1.a (map with IGR shows poor resolution) and 1.c (mass photometry table), Extended Data Figure 2.e (map with domains and mutations with very poor resolution), Extended Data Figure 4.b (the image for the amino acid sequence shows better quality but the numbers on the left have been cut), Extended Data Figure 5d the top picture of chains 3A-3Q is cropped, Extended Data Figure 6.c (mass photometry table) and 6.e (β -gal activity graph).

Extended Data Fig.1.a: EMSA gels missing the Bound (B) and Free (F) labels

In Figure 3a and Extended Data Fig.3A in the legends, the panel letters don't match the type (lowercase/uppercase) with the rest of the panel letter.

Results, page 5, line 27-28 "indicating that Stl2 effectively inhibited Sak80a-mediated process (Figure 4C)." Is this referring to Figure 4D?

Extended data 6, panel f. Is the explanation missing from the legend?

Reviewer #2

(Remarks to the Author)

In this manuscript, the authors attempt to characterize an anti-phage defense that is based on the sensing and regulatory mechanisms of a type of phage satellite, SaPI. In the proposed mechanism, stl encoded by SaPI2 inhibits phage-encoded homologous recombination proteins.

Although I find the idea interesting, and I appreciate some of the results of the mechanistic interaction of stl with Sak and Sak4, I was not convinced on (or clarified of) the effect of stl in providing protection against phage. SaPIs have other genes that are also not typically considered as "anti-phage defenses", but that contribute to a severe decrease in phage replication. For instance the ppi (phage packaging interference) and cpm (capsid morphogenesis) genes have been shown to interfere strongly with the production of phage PFUs. Importantly, SaPI2 has these proteins, and their effect in limiting phage replication has been quantified before (see for instance Table 1 in Ram et al, PNAS, 2014 (10.1073/pnas.1204615109)). Thus, I feel that more data is needed to show that stl, by itself, protects against phage infection. My reasoning for such is based on the points below:

1) First of all, I found some of the statements of novelty throughout the paper to be a bit forced. There is an important paper that, as far as I could tell, is not cited or discussed, but that follows closely the approaches and results of this work. In Rafael Ciges-Tomas, Nat Comms 2019 (10.1038/s41467-019-11504-2), the authors resolve the structure of the stl of SaPIbov1, showing its ability to form dimers and be derepressed by different phages. It is important to discuss the results here in light of this work.

2) Related to my main concern (the effect of stl for defense against phage infection), the authors describe that stl is a master repressor, which is consistent with the literature. However, often SaPIs have multiple repressors, the repression might not be complete, and some functions of the SaPI might actually interact with stl in a more complex way beyond repression. Since the focus of this paper is on the role of stl for phage inhibition, it would be important to verify that in all cases (meaning, all SaPIs tested, see point 3 below), all the other SaPI genes are indeed constitutively expressed in the absence of stl, as stated by the authors (page 2, lines 31-32). This would help to validate the hypothesis that stl by itself provides protection against phage.

3) Unless I missed it, the paper does not show data for the effect of a SaPI2 stl-deleted mutant. Figure 1B has the stl-deleted mutants of SaPI1 and SaPIbov1, but it is strangely missing SaPI2, whose stl2 is the focus of the paper. Although it is tempting to assume the effect would be the same for stl2 in SaPI2, it is necessary that is shown (as well as the data from point #2 above). Later on, the authors show the data for a SaPI2 with the stl2-sheetCTD variant, but I would argue that it is not the same, since the latter means that the Sak cannot interact with stl2 as before, if I understood correctly.

4) Moreover, the opposite experiment, that would show that stl2 is sufficient to provide protection against phage infection (without the action of other SaPI genes) is also missing. The inclusion of stl2 in a plasmid, and in the absence of the satellite, and the subsequent quantification of the protection in the presence/absence of the plasmid is essential to support the idea that stl is by itself a key and novel anti-phage defense mechanism.

5) My knowledge regarding cryo-EM and structural-based analyses is admittedly very limited, but based on the results shown here I could see that stl might target the Sak and Sak4 structures. And I do find interesting the fact that multimeric organizations of stl could explain the detection of different structures (and thus of different putative helper phages). What I am

less convinced about (and going back to my main criticism), is of the effect that this inhibition could have on the phage replication itself. Phage encoded HRs are typically used for genetic exchanges that drive the mosaicism of phage genomes (e.g. De Paepe et al, PLoS Genetics, 2014 (10.1371/journal.pgen.1004181)). Although I could see that this is a protein that could be used by SaPI to detect incoming phages, why would its inhibition, by itself, limit phage replication? This is not obvious to me. Perhaps the authors could expand a bit on possible mechanisms for this?

I have also other remarks, that although less critical I would like to see addressed:

- Going back to somewhat (but less important) exaggerated novelties, it is not immediately clear to me why spot assays would result in different results from those in standard plaque assays where PFUs are counted. But I am interested in the views of the authors on this. Is it because of a potential higher phage MOI? Moreover, the idea that phage-parasites (i.e., satellites) are themselves mechanisms of protection against phages is not exactly novel. For instance, PLEs in *Vibrio* are extremely efficient (population-wise) protectors against infection by the phage ICP1. In some circumstances, even P4 can lead to a severe decrease in the production of particles that carry the P2 phage.

- In page 7, lines 3-4, "... suggest that the ability to oligomerize has emerged also has a necessary condition for the induction mechanism of SaPI2, which is a unique feature among SaPIs". Do the authors have a reference or supporting data for this? According to the paper I already mentioned (Rafael Ciges-Tomas, Nat Comms 2019), at least the StI of SaPIbov1 has the ability to form dimers for its interaction with the phage, which is part of the reason for the promiscuity of this satellite.

- Assuming that stI does provide protection against the helper phage of the satellite, I would have liked to see some discussion on whether this could be advantageous or detrimental for the satellite. This also goes back to my main points above. Some satellites that encode anti-phage defenses (P4, PIC1) tend to decrease the infection by phages that are NOT their helpers, and hence cannot be exploited by the satellites. To my knowledge, the genes in all types of satellites (including in SaPI) that are known to interfere with the replication of their HELPER phages actually have a role in the hijacking itself (packaging interference, capsid modifications), meaning that the reduction in phage particles that carry the phage leads to a potential increase of particles that carry the satellite. However, if we do consider that an inhibition of phage replication by stI does occur, as stated by the authors, it is not clear that this would be beneficial to the satellite itself. If it decreases the replication of the phage, it could lead to less particles that are hijackable by the satellite, and hence be detrimental to its transfer. Although it could be argued that this also occurs with PLEs in *Vibrio* (which cause abortive infection of the cell before the virulent phages can properly replicate), their dynamics are peculiar and tend to differ from the other satellites. So perhaps the authors could expand a bit on this, in the discussion of the results?

- As I mentioned before, I am not an expert in biochemistry or structural analyses. Thus, maybe it was mostly for me, but a large part of the paper is a bit dense in terms of terminology. Given that Nat Comms is a generalist journal, I would suggest some work in making the text more accessible to a general audience.

- Some figures/panels have a really poor quality, at least in my version (e.g., Fig4B, table in Fig4C and ED Fig8)

Reviewer #3

(Remarks to the Author)

In "Phage-parasites targeting phage homologous recombinases provide robust antiviral immunity", the authors describe novel cryo-EM and biochemical results on the phage-parasite SaPI2. I was asked to review the structural analysis of this work, so this review will be limited to the structural aspects. In this manuscript the authors have determined an impressive number of very unique structures. The cryo-EM structure determination is of high quality, and their structures are convincing. The structure of Sak80 in complex with StI2 is the crux of the paper and is really an extraordinary structure with two Sak80 α rings held together by a split ring 15mer of StI2. Based on this structure, the authors propose a very plausible model for the mechanism of inhibition employed by SaPI2. The main issue I have with the manuscript is that it is quite hard to follow. In particular, it could really benefit from a schematic model for the proposed mechanisms of the different complexes. However, the structures described are sound and quite novel.

A few other specific concerns are listed below

Why is the term D1 symmetry used? Should it not be C2?

For this statement: "We generated a cryo-EM map of the undecamer state with C11 symmetry (structure not presented here) and discovered that the circles were formed by 11, 12 and 13 dimers of StI2." This data should be shown. How did you discover the different forms? Also, a minor point is that this sentence is redundant with the previous one.

Page 1, line 34 "which in addition to repress" should be "which in addition to repressing"

The methods section needs substantial copy editing

Kryos -> Krios for all instances

Reviewer #4

(Remarks to the Author)

Staphylococcus pathogenicity islands (SaPI) are known to inhibit certain bacteriophage that prey upon staphylococci. To gain mechanistic insight into this phenomenon, Debiasi-Anders and co-workers generated an *stl* deficient *S. aureus* to constitutively activate SaPI2. They surprisingly found that *stl* deficient *S. aureus* had become sensitive to phage 80 α suggesting that the *Stl* proteins may be responsible for phage inhibition. The investigators recombinantly expressed and purified *Stl2*. Its previously known activity as a transcriptional repressor that binds to a specific intergenic region was confirmed. Analysis of its oligomeric state revealed, in addition to its anticipated dimeric form, a higher order oligomer. Cryo-EM analysis of this structure produced class averages of ring structures and fiber structures. Single particle reconstructions of apo *Stl2* were generated which depict how the dimer forms and the residues that mediate dimer formation into higher order structures (rings and filaments). *Stl2*-sheetCTD mutants demonstrated that dimers incapable of forming higher order structure can still DNA binding: the transcriptional regulation functions of *Stl2* do not require the rings/filaments. This was further supported by *in vivo* reporter assays.

The investigators turned to a potential interaction of *Stl2* with bacteriophage homologous recombinases. The structure of an N-terminal fragment Sak80 α was determined by cryo-EM showing it adopted an oligomeric ring structure which was expected. ssDNA is expected to bind to the outside surface of the ring based on homologous structures. The investigators showed incubation of *Stl2* and Sak80 α leads to the formation of higher-order structures and the inhibition of the DNA binding and annealase activity of the two proteins respectively. The cryo-EM structure of the *Stl2*- Sak80 α complex was solved and this revealed that *Stl2* forms a collar around the Sak80 α . The authors hypothesize that in this arrangement *Stl2* blocks the entry of ssDNA into the ssDNA binding groove of Sak80 α , although other mechanisms for the inhibition of Sak80 α activity are possible. To test whether *Stl2* can form ultra-structures with other phage homologous recombinases a structure of *Stl2* with Sak452A was solved.

An overall strength of this investigation is the array of complementary techniques used: *in vivo* phage sensitivity assays, *in vitro* biochemistry, *in vivo* transcriptional reporter assays and structural biology. The conclusions appear well supported by the data. The methods reporting is detailed and the mechanistic conclusions are impactful for the field of phage biology.

Questions/Comments:

1. Is the *Stl2* apo dimer reconstructed from the ring-shaped particles or all particles?
2. In the last Results paragraph it is stated "it seems reasonable to presume that *Stl2* homologs will be eventually found spanning all domains of life." However, the *in-silico* analysis presented only finds *Stl2* homologs in eubacteria so altering this statement to "spanning many eubacteria" seems better suited.

Typos etc.

Page 4 line 28 should be a callout to Figure 4D not Figure 4C.

Page 7 line 9 should be "supporting" the hypothesis rather than "proving" it.

Page 7 line 46 would read better altered to "Using the standard spots assay.."

Page 12 line 22 BL21-(DE3)

Page 13 line 11 Sanger (capitalized)

Version 1:

Reviewer comments:

Reviewer #1

(Remarks to the Author)

Thank you for the opportunity to review the revised manuscript of Debiasi-Anders et al. The authors have performed additional experiments that have clarified some of the questions raised in the previous revision. Overall, the work is substantial, and I am happy to recommend this revised version to be accepted for publication.

Reviewer #2

(Remarks to the Author)

I believe this version of the manuscript improved the previous one, and addressed many of my criticisms. Indeed, my previous review focused on the independent function of *stl2* in providing immunity. This stems not from how I perceive the system, but how the authors wrote their findings in the previous version of the manuscript. Sentences like "we present the inhibitory mechanism of *Stl2* against Sak80 α ", suggest this self-sufficiency of *stl2* in this immunity. This is what I argued that was not convincingly shown in the previous manuscript, and that the authors showed in the reviewed version that was indeed not the case. It makes more sense that it is an *Stl2*-mediated mechanism, and that does not diminish the interest of the results. But it is important to be clear on how the role of *Stl2* is communicated.

I appreciate also how more information and references were included on how the targeting of HR mechanisms can alter the phage infectiveness/reproductive fitness. And I appreciate as well the discussion of the impact for such mechanisms in the SaPI-helper interactions.

I have only a few minor points that I would like the authors to address.

I find it intriguing that *stl2* mutants are not viable in SaPI2, whilst that is not the case in the other SaPIs tested. Of course the reason why is not the point of this study, but perhaps the authors could speculate on why this is the case. Could the absence

of stl2 provide aggressive auto-immunity, that is not as severe in the case of overloading stl2 with sak80a vector?

Page 2, lines 28-29: "The effect that SaPIs might have during phage infection has not been studied using spot assays." -> It is still not clear why one would expect it to be different. My hypothesis would be that bacteria could express different receptors in liquid or solid, but this is not stated, nor a reference for an example or dynamics of infection being different between liquid and solid environments is included. Which brings me to the question (which maybe I missed), is the effect of stl2-mediated inhibition of the phage observed also in liquid media, or only in plaques? This wouldn't change (for me) the impact of the result, but it might provide more information regarding the ecological impact of the stl2-mediated inhibition mechanism.

Page 8, Lines 38-39: "may contribute to explain the widespread presence of these phage-parasites in virtually any microbiome on earth" -> Although satellites have been shown to be well spread amongst different species of bacteria, as far as I am aware, that they are present in "virtually any microbiome" has not been systematically demonstrated (using, e.g., metagenomic data). Please add a reference (in the case I am missing something) or adjust the sentence accordingly.

Finally, I think the authors missed my point about other satellites ACTING as (and not encoding) defense systems. When I mentioned PLE, it is because PLE itself does not have accessory genes providing defense systems, but the PLE itself IS itself the defense system, as its hijacking mechanism prevents the phage from replicating and then kills the cell. To me, this is different from the satellite-encoding defense systems (like the P4s and their systems in the Rousset et al reference), since, again, these are accessory genes, and not part of the mechanistic functions of hijacking of the satellite itself. And I believe this is what the authors are proposing here as well, by saying that SaPI2 does not encode other (accessory) defense systems, but that it is the mechanism of hijacking itself that protects against the phage. So it would be nice to put this in context with the other satellites that do the same.

REVIEWER COMMENTS

Dear Editor, we thank all four reviewers for their great suggestions. We have answered point by point as concise as possible. We apologize for the delay in our response due to the flooding in Valencia. Since I was born and raised there, the catastrophic events had quite an impact in my schedule and productivity. We are very thankful for your understanding with the submission's deadline.

Within the manuscript, all new text changes are in blue.

The revised figures:

- Revised figure 1, we have included a new experiment that mimics a *stl2* mutant (Figure 1B). Because the lysates used in these experiments were new, we have used again RN4220, SaPI2 containing RN4220 and empty vector as controls in these replicates. We include also a nitrocefin assay (see answer to reviewer #2) to confirm that the phage interference gene *ppi* is expressed in a *stl* mutant. We have removed phage SLT in figure 1C because we do not characterize Erf recombinase in this paper, and we think it might distract the readers.
- Revised Extended Data Figure 1F, where we include the map and model of the D11 circles as requested by reviewer #3.
- New figure 7, with a schematic model of the process (reviewer #3).
- The changes in the text of the figure legends are not highlighted in blue.

In our view, this revised version has improved substantially. We summarize the findings of our work as follows: Phage-parasites provide immunity against phage infection. Such immunity is mediated by Stls, whose function is necessary but not sufficient when expressed from a vector. When the parasite targets the phage homologous recombinases via Stl2, the immunity is astonishingly robust. Stl2 mode of action is unique respect other Stls. Stl2 represents a remarkable evolutionary adaptation.

We look forward to hearing from you with a decision.

All the best,

Ignacio Mir Sanchis on behalf of the authors.

Reviewer #1 (Remarks to the Author):

In this study, Debiasi-Anders et al. have characterized to the structural level the interaction between the SaPI2 repressor protein, Stl2 with two different families of recombinases, Rad52-like Sak80 α and Rad51-like Sak452A. The authors have used a wide range of biochemistry techniques, in vivo analysis and they have produced structural data that provide strong evidence of these interactions. Since the discovery of these phage parasites and specially in the last 10 years, much work has been produced to elucidate the molecular interactions between the SaPIs and their helper phages. The work presented in this manuscript adds up to that knowledge by elucidating how SaPI2 interfere with their targeted phage. The detailed structural data provided in this work by solving the structure of the Stl2 repressor, shows it oligomerize as rings/filaments to mimic the folding of their targets: Sak80 α (rings) and Sak452A (filaments). The authors

further confirm these interactions by solving the complexed structures of the Stl2 with Sak80 α and Sak452A. The authors also show that these interactions impact the helper phage ability to infect and propose this mechanism as another way SaPIs can provide immunity without encoding canonical immune systems. Given the high interest in the present time about the study of the bacteria immunity systems arsenal, this work could be of potential interest to the microbiology community. Overall, the manuscript is well-written, the rationale for each experiment is well laid out and the results are sound. There are few comments for the authors to be addressed before the manuscript could be accepted for publication:

Abstract, page 1, lines 13-15: “Unrelated anti-phage defense systems target phage-encoded homologous recombinases (HRs). The inhibition mechanisms of anti-phage systems targeting phage encoded HRs remain uncharacterized.”

Which are these anti-phage defense systems targeting HRs? Other than in the Abstract there is no mention to other anti-phage defense systems targeting HRs through the text.

We thank the reviewer for this review. We had to remove the references from the abstract, where they were included in previous versions of the manuscript. Our apologies for that.

In addition to the SaPIs, these systems are i) abortive infection mechanisms in *Lactococcus* (Bouchard et al 2004), ii) a serine/threonine kinase in *Staphylococcus* (Depardieu, 2016), iii) CRISPR (Steczkiwicz et al, 2021)

We have now included a paragraph in the introduction:

*Given the importance of homologous recombination it is not surprising that several lines of evidence indicate that unrelated phage defense systems target phage-encoded HRs spanning all superfamilies. The study of plasmid-encoded Abortive infection mechanisms (Abi) led to the identification and initial characterization of proteins in *Lactococcus lactis* bacteriophages named Sak (Sensitivity to AbiK) (Bouchard et al 2004). A few years later it was discovered that an Abi system in *Staphylococcus aureus* also targets phage recombinases (Depardieu, 2016). Recently, it has been found CRISPR spacers in Firmicutes matching single strand annealing protein genes (ssap genes) indicating that CRISPR systems also target phage annealase (Steczkiwicz et al, 2021)*

Results, page 2, lines 30-31: “To investigate this, we utilized variants of SaPI1 and SaPIbov1 in which the stl genes have been inactivated through genomic mutagenesis: SaPI1 Δ stl, SaPIbov1 Δ stl.”

It is strange to me why the authors following the rationale for this experiment did not include SaPI2 Δ stl in this experiment. The authors hypothesized that the strong interference phenotype for SaPI2 is due to Stl2, but without testing the actual Stl2 mutant to corroborate their hypothesis.

We agree with the reviewer that this mutant is an important and logical point in the story. We have tried to generate the *stl2* mutant, either by deletion or insertion of a resistance cassette, in both cases employing the pMAD vector for allelic replacement. Unfortunately, we have not been able to generate it. When confirming potential mutants, we always get two bands, one band corresponding to the mutant and another band corresponding to the wild-type. Although we have tested these potential mutants and they are also sensitive to phage 80 α infection, we are not

convinced that they are clean mutants and we do not feel comfortable using them and claiming that we have generated a bona fide *stl2* mutant.

To overcome this limitation, we have done an extra experiment in which we try to mimic the *stl2* mutant conditions. In this experiment we ectopically express *sak_{80α}* from the pGTM3 expression vector (cadmium inducible promoter P_{cad}) to force the release of Stl2 from its operator, then we infect with phage 80α. As seen in the new Figure 1B, there is not Stl2-mediated protection and cells become sensitive to phage 80α infections as it was the case for SaPI1 and SaPI_{bov} *stl* mutants. This experiment shows that in a complete physiological scenario, when Stl2 is removed from the equation, the immunity is lost even when other phage interference molecules are expressed (see *ppi* transcriptionally fusion experiment below).

Results, page 2, lines 34-38: “Surprisingly, we discovered that the absence of the *stl* gene in SaPI1Δ*stl* and SaPI_{bov}1Δ*stl* variants turned them sensitive to phage infection, indicating that constitutive expression of SaPI-encoded genes in these variants was not sufficient for phage protection. Moreover, this result suggested the necessity of *stl* for full immunity (Figure 1B).” This result is indeed surprising, since it is known that these SaPIs although not carrying “canonical” anti-phage defense systems, they do carry different factors to interfere with the phage reproduction such as *ppi* (interfere with helper phage DNA packaging), *cpmA* and *cpmB* (capsid size redirection genes) or *ptiA*, *ptiB* and *ptiM* (interference with helper phage late gene expression). Do the authors have a hypothesis as to why in a mutant in *stl*, where these factors should be constitutively expressed, they do not provide immunity?

We have speculated a bit in our response to reviewer #2’s point 4 and in a new paragraph in the discussion. Phage-parasites need to interfere with the helper phage in a balanced way, not too tight, not too loose, since the phage reproduction is also important for the parasite’s success to be disseminated. It looks like these interference systems need to be also finely expressed to interfere in a balanced way. Eutopic expression of these systems seems to reflect this feature.

Results, page 4, lines 21-22: “We struggled to obtain a good quality cryo-EM map of full length Sak80a, consequently, we decided to focus on a truncated version of Sak80a where its CTD was removed”

It would be important to explain the rationale behind the decision to remove the CTD domain, given that later it is shown as the part that interacts with Stl2.

We thank the reviewer to point this out. We should have mentioned it, indeed. It is known that removing the CTD helps to get a data set without preferred oriented particles.

We have included an explanation in the text: *The full length dataset presented two major challenges: severe particle orientation preference and the absence of the flexible CTD, which is a common issue with Rad52-like annealases.*

Results, page 4, line 27: what does SSAPs stands for?

The discussion is quite concise, but in my opinion, it was missing some thoughts about the Gp2.5-like family of helicases that also interacts with Stl2. Do the authors have any hypothesis as to which kind of interactions between Stl2 and this family of helicases could be occurring? Would Stl2 be also mimicking the structure of Gp2.5-like helicases?

We do not have a hypothesis about the interaction's nature between Stl2 and Gp2.5-like although it is under experimental investigation in our lab.

Methods, page 18, line 28-29: "All maps and models have been deposited in public databases." Please include in this section the PDB ID for the structures deposited in this manuscript.

Done

Minor comments:

There are some images with poor quality or poor resolution when zoomed in: Figure 1.A (maps' SaPIs with IGR), Figure 4.B (β -gal activity graph) and 4.C (mass photometry table), Extended Data Figure 1.a (map with IGR shows poor resolution) and 1c (mass photometry table), Extended Data Figure 2.e (map with domains and mutations with very poor resolution), Extended Data Figure 4.b (the image for the amino acid sequence shows better quality but the numbers on the left have been cut), Extended Data Figure 5d the top picture of chains 3A-3Q is cropped, Extended Data Figure 6.c (mass photometry table) and 6.e (β -gal activity graph).

Thanks! We did not include high resolution images for the initial submission. We think it was Adobe while creating pdf files with reduced size.

Extended Data Fig.1.a: EMSA gels missing the Bond (B) and Free (F) labels

Done

In Figure 3a and Extended Data Fig.3A in the legends, the panel letters don't match the type (lowercase/uppercase) with the rest of the panel letter.

Done

Results, page 5, line 27-28 "indicating that Stl2 effectively inhibited Sak80a-mediated process (Figure 4C)." Is this referring to Figure 4D?

Changed

Extended data 6, panel f. Is the explanation missing from the legend?

Thanks for catching this one! Fixed.

Reviewer #2 (Remarks to the Author):

In this manuscript, the authors attempt to characterize an anti-phage defense that is based on the sensing and regulatory mechanisms of a type of phage satellite, SaPI. In the proposed mechanism, stl encoded by SaPI2 inhibits phage-encoded homologous recombination proteins.

Although I find the idea interesting, and I appreciate some of the results of the mechanistic interaction of stl with Sak and Sak4, I was not convinced on (or clarified of) the effect of stl in

providing protection against phage. SaPIs have other genes that are also not typically considered as "anti-phage defenses", but that contribute to a severe decrease in phage replication. For instance the *ppi* (phage packaging interference) and *cpm* (capsid morphogenesis) genes have been shown to interfere strongly with the production of phage PFUs. Importantly, SaPI2 has these proteins, and their effect in limiting phage replication has been quantified before (see for instance Table 1 in Ram et al, PNAS, 2014 (10.1073/pnas.1204615109)). Thus, I feel that more data is needed to show that *stl*, by itself, protects against phage infection. My reasoning for such is based on the points below:

We thank the reviewer for this great review. The main point of this review lays on the fact that *Stl2* should function independently (by itself, as it has been repeated many times throughout the review) to be considered involved in immunity, in other words, that it should be sufficient to provide immunity. We do not fully agree with this statement, since molecules might be sufficient, might be necessary or might be sufficient and necessary regarding their functions. We thank the reviewer to prompt us to elucidate this matter. After doing the suggested experiments, we say in this revised version that *Stl2* is necessary but not sufficient regarding the immunity when it is expressed from a cadmium inducible promoter *P_{cad}*. We address the reviewer's concerns that *ppi* is expressed in the absence of *Stl2* and *Stl_{bov}*, demonstrating that SaPI-encoded proteins directly involved in phage reproduction's interference are not sufficient either, as it is for the *Stls*, to provide immunity in a complete physiological scenario.

1) First of all, I found some of the statements of novelty throughout the paper to be a bit forced. There is an important paper that, as far as I could tell, is not cited or discussed, but that follows closely the approaches and results of this work. In Rafael Ciges-Tomas, Nat Comms 2019 (10.1038/s41467-019-11504-2), the authors resolve the structure of the *stl* of SaPI_{bov1}, showing its ability to form dimers and be derepressed by different phages. It is important to discuss the results here in light of this work.

We have included a comment in the discussion where we highlight similarities and differences about the mode of action of different *Stls*. The following text has been added:

The mode of action of Stl2 is distinct from that of the two other characterized Stls (Stl-bov1 and Stl1) and from distant repressors found in other mobile genetic elements. Dimeric Stl-bov1, for instance, targets phage dUTPases that fold as dimers or trimers and dissociates into monomers upon target binding³⁴. In contrast, the tetrameric form of Stl1 appears to undergo a conformational change in its DNA binding domain, rendering it incompatible with operator binding³⁵. To the best of our knowledge, Stl2's ability to oligomerize into large multimers represents a novel mode of action previously unobserved. In our view, this represents a remarkable example of evolutionary structural-functional adaptation.

2) Related to my main concern (the effect of *stl* for defense against phage infection), the authors describe that *stl* is a master repressor, which is consistent with the literature. However, often SaPIs have multiple repressors, the repression might not be complete, and some functions of the SaPI might actually interact with *stl* in a more complex way beyond repression. Since the focus of this paper is on the role of *stl* for phage inhibition, it would be important to verify that in all cases (meaning, all SaPIs tested, see point 3 below), all the other SaPI genes are indeed

constitutively expressed in the absence of *stl*, as stated by the authors (page 2, lines 31-32). This would help to validate the hypothesis that *stl* by itself provides protection against phage.

It is well established that mutations in the *stl* genes of SaPI_{bov1} and SaPI₁ lead to the uncontrolled expression of the *xis* gene as well as the replication genes, enabling autonomous replication of the islands (Ubeda et al., 2008). These mutants exhibit a distinctive phenotype characterized by slower growth and a yellow pigmentation of colonies in RN4220. Notably, RN4220 is typically unpigmented due to a mutation in *rsbU*. However, under stress conditions, such as exposure to subinhibitory concentrations of antibiotics (Herbert et al., 2001), pigmentation is restored. This suggests that the stress induced by the uncontrolled replication of these SaPI mutants is responsible for both the pigmentation and slow growth. It is also generally assumed, (including reviewer #1) that the other genes (*ppi*, *cpmA*, *cpmB*) are also expressed. To address reviewer #2's concerns and verify this assumption, we transcriptionally fused the *ppi* gene to the β -lactamase reporter gene present in the pCN41 vector. This allowed us to measure *ppi* expression in both wildtype (*stl* present) and mutant (*stl* absent) constructs. We generated these constructs for SaPI_{bov1} and SaPI₂ islands but were unable to do so for SaPI₁. As shown in new Figure 1B, right-hand side panel, the presence of *stl2* and *stlbov1* inhibited the expression of the *ppi*-fused reporter gene, while in the absence of *stl2* and *stlbov1*, reporter expression was observed, confirming that the *ppi* gene is indeed expressed in *stl* mutants. We do not quite understand why we could not generate the SaPI₁ construct. It is known that plasmids with two origins of replications are unstable. Within these constructs, the SaPI-encoded origin of replication is included (it is located upstream the *ppi* genes). We hypothesized that for some unknown reason, the SaPI₁'s origin of replication but not those from SaPI₂ and SaPI_{bov1} were interfering with the cloning process. We then generated the remaining SaPI₁ constructs (*stl1* WT and *stl1* mutant) where the origin of replication was removed by overlapping PCRs. Using this strategy, we were able to generate the constructs for SaPI₁ and measure the expression of the SaPI₁-encoded *ppi* gene by the nitrocefin assay. Again, the presence of the *stl1* prevented the expression of SaPI₁-encoded *ppi*, while in the absence of *stl1* the expression of the *ppi* gene occurred.

3) Unless I missed it, the paper does not show data for the effect of a SaPI₂ *stl*-deleted mutant. Figure 1B has the *stl*-deleted mutants of SaPI₁ and SaPI_{bov1}, but it is strangely missing SaPI₂, whose *stl2* is the focus of the paper. Although it is tempting to assume the effect would be the same for *stl2* in SaPI₂, it is necessary that is shown (as well as the data from point #2 above). Later on, the authors show the data for a SaPI₂ with the *slt2*-sheetCTD variant, but I would argue that it is not the same, since the latter means that the Sak cannot interact with *slt2* as before, if I understood correctly.

We reply to this question in our answer for reviewer #1's second point

4) Moreover, the opposite experiment, that would show that *stl2* is sufficient to provide protection against phage infection (without the action of other SaPI genes) is also missing. The inclusion of *stl2* in a plasmid, and in the absence of the satellite, and the subsequent quantification of the protection in the presence/absence of the plasmid is essential to support the idea that *stl* is by itself a key and novel anti-phage defense mechanism.

We thank the reviewer for suggesting this experiment. We have done the experiment and discovered that when expressed from the exogenous Pcad, Stl2 is not sufficient to provide immunity against phage 80 α (not shown). We are happy to show the plates of the experiment if requested. As mentioned above, we do not fully agree with the statement that a molecule must be sufficient to be considered a key player in a particular function. As reviewer #1 pointed, three different mechanisms have been identified to interfere with phage reproduction: *ppi*, *cpmAB* and *ptiBMA* (Ubeda et al 2009, Damle et al 2012, Ram et al 2012 and Ram et al 2014). The first two mechanisms are effective against phage 80 α , so is the third one against phage 80. We would like to mention that a SaPI2 double mutant in *ppi* and *cpmAB* still shows a remarkable resistance against phage 80 α . The size of the plaques in figure 3B bottom panel (Ram et al 2012) of the double mutants indicate that the phage reproduction is severely compromised when compared to no SaPI, indicating that there must be a fourth mechanism regarding the phage 80 α . Similarly, in figure 3B top panel, SaPIbov1 double mutant *ppi* and *cpmAB* is strikingly resistant. Finally, figure 4 in the same reference, SaPI2 delta orf17 is almost completely resistant against phage 80 (note *ppi* and *cpmAB* have no effect on phage 80). In summary if all the SaPI-encoded interference mechanism are removed through mutagenesis, the cells remain strikingly resistant.

In our view, our findings that the cells become completely sensitive in the absence of *stls* indicate that Stls are such fourth mechanism. Why is the phage so 'healthy' in a *stl* defective scenario? We were expecting to see the phage suffering. We agree that it is puzzling. We cannot explain it yet. We suspect that the evolutionary arms race between phages and SaPIs is so fierce that as soon as one touches anything on the SaPI side, the phage wins. This might be the reason why SaPI2 encodes several mechanisms against phages which are seen as redundant. As mentioned by this reviewer later, and discussed many times in the field, the satellite needs to interfere with the helper phage but not too much so that it is not detrimental for the satellite itself.

5) My knowledge regarding cryo-EM and structural-based analyses is admittedly very limited, but based on the results shown here I could see that *stl* might target the Sak and Sak4 structures. And I do find interesting the fact that multimeric organizations of *stl* could explain the detection of different structures (and thus of different putative helper phages). What I am less convinced about (and going back to my main criticism), is of the effect that this inhibition could have on the phage replication itself. Phage encoded HRs are typically used for genetic exchanges that drive the mosaicism of phage genomes (e.g. De Paepe et al, PloS Genetics, 2014 (10.1371/journal.pgen.1004181)). Although I could see that this is a protein that could be used by SaPI to detect incoming phages, why would its inhibition, by itself, limit phage replication? This is not obvious to me. Perhaps the authors could expand a bit on possible mechanisms for this?

The reviewer is right by mentioning that phage recombinases drive phage genome mosaicism. However, it was demonstrated in Neamah et al 2017 [10.1093/nar/gkx308](https://doi.org/10.1093/nar/gkx308) that these recombinases were involved in phage replication. It was proposed that these recombinases, in staphylococcal phages at least, switch from the theta mode to the sigma mode of DNA replication.

Note that mutants in *sak* and *sak4* completely impair phage reproduction. Replication is vastly decreased, and no plaques are generated at all. This is why we reasoned that Stl2 targeting phage

HR would compromise the DNA replication of the phage.

I have also other remarks, that although less critical I would like to see addressed:

- Going back to somewhat (but less important) exaggerated novelties, it is not immediately clear to me why spot assays would result in different results from those in standard plaque assays where PFUs are counted. But I am interested in the views of the authors on this. Is it because of a potential higher phage MOI? Moreover, the idea that phage-parasites (i.e., satellites) are themselves mechanisms of protection against phages is not exactly novel. For instance, PLEs in *Vibrio* are extremely efficient (population-wise) protectors against infection by the phage ICP1. In some circumstances, even P4 can lead to a severe decrease in the production of particles that carry the P2 phage.

Yes, we agree that other phage satellites protect against phages. Quite diverse number of defense mechanisms have been identified in satellites as we have mentioned and cited:

Rousset, F., et al (2022). *Phages and their satellites encode hotspots of antiviral systems*. *Cell Host & Microbe* 30, 740-753.e5. <https://doi.org/10.1016/j.chom.2022.02.018>.

We were surprised of the level of protection that we saw in the spot assays and are investigating possible explanations.

- In page 7, lines 3-4, "... suggest that the ability to oligomerize has emerged also has a necessary condition for the induction mechanism of SaPI2, which is a unique feature among SaPIs". Do the authors have a reference or supporting data for this? According to the paper I already mentioned (Rafael Ciges-Tomas, *Nat Comms* 2019), at least the StI of SaPI_{bov1} has the ability to form dimers for its interaction with the phage, which is part of the reason for the promiscuity of this satellite.

See our answer to point 1. We have reworded the text to avoid confusion. We did not refer to StI2 making dimers, but oligomerizing as a larger multimers.

- Assuming that stl does provide protection against the helper phage of the satellite, I would have liked to see some discussion on whether this could be advantageous or detrimental for the satellite. This also goes back to my main points above. Some satellites that encode anti-phage defenses (P4, PICI) tend to decrease the infection by phages that are NOT their helpers, and hence cannot be exploited by the satellites. To my knowledge, the genes in all types of satellites (including in SaPI) that are known to interfere with the replication of their HELPER phages actually have a role in the hijacking itself (packaging interference, capsid modifications), meaning that the reduction in phage particles that carry the phage leads to a potential increase of particles that carry the satellite. However, if we do consider that an inhibition of phage replication by stl does occur, as stated by the authors, it is not clear that this would be beneficial to the satellite itself. If it decreases the replication of the phage, it could lead to less particles that are hijackable by the satellite, and hence be detrimental to its transfer. Although it could be argued that this also occurs with PLEs in *Vibrio* (which cause abortive infection of the cell before the virulent phages can properly replicate), their dynamics are peculiar and tend to differ from the other satellites. So perhaps the authors could expand a bit on this, in the discussion of

the results?

We have included a paragraph in the discussion to speculate about this.

- As I mentioned before, I am not an expert in biochemistry or structural analyses. Thus, maybe it was mostly for me, but a large part of the paper is a bit dense in terms of terminology. Given that Nat Comms is a generalist journal, I would suggest some work in making the text more accessible to a general audience.

We have tried to change some sentences here and there to make it more general. All changes are marked in blue.

- Some figures/panels have a really poor quality, at least in my version (e.g., Fig4B, table in Fig4C and ED Fig8).

Fixed

Reviewer #3 (Remarks to the Author):

In “Phage-parasites targeting phage homologous recombinases provide robust antiviral immunity”, the authors describe novel cryo-EM and biochemical results on the phage-parasite SaPI2. I was asked to review the structural analysis of this work, so this review will be limited to the structural aspects. In this manuscript the authors have determined an impressive number of very unique structures. The cryo-EM structure determination is of high quality, and their structures are convincing. The structure of Sak80 in complex with Stl2 is the crux of the paper and is really an extraordinary structure with two Sak80 α rings held together by a split ring 15mer of Stl2. Based on this structure, the authors propose a very plausible model for the mechanism of inhibition employed by SaPI2. The main issue I have with the manuscript is that it is quite hard to follow. In particular, it could really benefit from a schematic model for the proposed mechanisms of the different complexes. However, the structures described are sound and quite novel.

We thank the reviewer for this review. Schematic model has been included in new Figure 7

A few other specific concerns are listed below

Why is the term D1 symmetry used? Should it not be C2?

We used the term D1 because we imposed D1 symmetry.

For this statement: “We generated a cryo-EM map of the undecamer state with C11 symmetry (structure not presented here) and discovered that the circles were formed by 11, 12 and 13 dimers of Stl2.” This data should be shown. How did you discover the different forms? Also, a minor point is that this sentence is redundant with the previous one.

We thank the reviewer for this comment. We now realize that our initial text was confusing. We discovered 11, 12 and 13 protomers looking at the 2D classes. We then generated a map imposing C11 symmetry and realized that the Stl2 protomers were not monomers but dimers.

The dimers were formed in the dihedral axis. We generated a map imposing D11 symmetry which was a bit better than the C11 one. The information is not redundant albeit confusing. First, we mention 11, 12, 13 protomers but then we say that they are dimers. We have changed the wording to avoid confusion.

We did not show the map of the C11 nor the D11 circles because they are the same as in the deposited map that we have called three dimers. We have included a new panel in Extended Data Figure 1F with the map and model of the D11 circle.

Page 1, line 34 “which in addition to repress” should be “which in addition to repressing”

Changed

The methods section needs substantial copy editing

Kryos -> Krios for all instances.

Fixed

Reviewer #4 (Remarks to the Author):

Staphylococcus pathogenicity islands (SaPI) are known to inhibit certain bacteriophage that prey upon staphylococci. To gain mechanistic insight into this phenomenon, DeBiasi-Anders and co-workers generated an *stl* deficient *S. aureus* to constitutively activate SaPI2. They surprisingly found that *stl* deficient *S. aureus* had become sensitive to phage 80 α suggesting that the Stl proteins may be responsible for phage inhibition. The investigators recombinantly expressed and purified Stl2. Its previously known activity as a transcriptional repressor that binds to a specific intergenic region was confirmed. Analysis of its oligomeric state revealed, in addition to its anticipated dimeric form, a higher order oligomer. Cryo-EM analysis of this structure produced class averages of ring structures and fiber structures. Single particle reconstructions of apo Stl2 were generated which depict how the dimer forms and the residues that mediate dimer formation into higher order structures (rings and filaments). Stl2-sheetCTD mutants demonstrated that dimers incapable of forming higher order structure can still DNA binding: the transcriptional regulation functions of Stl2 do not require the rings/filaments. This was further supported by in vivo reporter assays.

The investigators turned to a potential interaction of Stl2 with bacteriophage homologous recombinases. The structure of an N-terminal fragment Sak80 α was determined by cryo-EM showing it adopted an oligomeric ring structure which was expected. ssDNA is expected to bind to the outside surface of the ring based on homologous structures. The investigators showed incubation of Stl2 and Sak80 α leads to the formation of higher-order structures and the inhibition of the DNA binding and annealase activity of the two proteins respectively. The cryo-EM structure of the Stl2- Sak80 α complex was solved and this revealed that Stl2 forms a collar around the Sak80 α . The authors hypothesize that in this arrangement Stl2 blocks the entry of ssDNA into the ssDNA binding groove of Sak80 α , although other mechanisms for the inhibition of Sak80 α activity are possible. To test whether Stl2 can form ultra-structures with other phage

homologous recombinases a structure of Stl2 with Sak452A was solved.

An overall strength of this investigation is the array of complementary techniques used: in vivo phage sensitivity assays, in vitro biochemistry, in vivo transcriptional reporter assays and structural biology. The conclusions appear well supported by the data. The methods reporting is detailed and the mechanistic conclusions are impactful for the field of phage biology.

Questions/Comments:

1. Is the Stl2 apo dimer reconstructed from the ring-shaped particles or all particles?

The Stl2 apo structure is reconstructed from all particles.

2. In the last Results paragraph it is stated “it seems reasonable to presume that Stl2 homologs will be eventually found spanning all domains of life.” However, the in-silico analysis presented only finds Stl2 homologs in eubacteria so altering this statement to “spanning many eubacteria” seems better suited.

Changed. We have included archaea because there are phage-parasite elements found in archaea as well.

Typos etc.

Page 4 line 28 should be a callout to Figure 4D not Figure 4C.

Done

Page 7 line 9 should be “supporting” the hypothesis rather than “proving” it.

Done

Page 7 line 46 would read better altered to “Using the standard spots assay..”

Done

Page 12 line 22 BL21-(DE3)

Done

Page 13 line 11 Sanger (capitalized)

Done

Dear Editor,
We have replied to the reviewers point by point in red.
Looking forward to hearing from you.
Best,
Ignacio Mir Sanchis.

2025-01-21

REVIEWERS' COMMENTS

Reviewer #1 (Remarks to the Author):

Thank you for the opportunity to review the revised manuscript of Debiasi-Anders et al. The authors have performed additional experiments that have clarified some of the questions raised in the previous revision. Overall, the work is substantial, and I am happy to recommend this revised version to be accepted for publication.

We appreciate the reviewer's recommendation.

Reviewer #2 (Remarks to the Author):

I believe this version of the manuscript improved the previous one, and addressed many of my criticisms. Indeed, my previous review focused on the independent function of stl2 in providing immunity. This stems not from how I perceive the system, but how the authors wrote their findings in the previous version of the manuscript. Sentences like "we present the inhibitory mechanism of Stl2 against Sak80a", suggest this self-sufficiency of stl2 in this immunity. This is what I argued that was not convincingly shown in the previous manuscript, and that the authors showed in the reviewed version that was indeed not the case. It makes more sense that it is an Stl2-mediated mechanism, and that does not diminish the interest of the results. But it is important to be clear on how the role of Stl2 is communicated.

We have reworded some sentences trying to communicate a bit better the role of Stl2, stressing the fact that is Stl2-mediated.

I appreciate also how more information and references were included on how the targeting of HR mechanisms can alter the phage infectiveness/reproductive fitness. And I appreciate as well the discussion of the impact for such mechanisms in the SaPI-helper interactions.

I have only a few minor points that I would like the authors to address.

I find it intriguing that stl2 mutants are not viable in SaPI2, whilst that is not the case in the other SaPIs tested. Of course the reason why is not the point of this study, but perhaps the authors could speculate on why this is the case. Could the absence of stl2 provide aggressive auto-immunity, that is not as severe in the case of overloading stl2 with sak80a vector?

The genetic manipulation of this SaPI locus has been always problematic. Attempting to make a double mutant integrase-stl is unviable in several SaPIs. Even cloning nearby segments of DNA into Ecoli is problematic. When we overload Stl2 by expressing sak80 α , the cells suffer as well. The reviewer's hypothesis could be the case. There might be unidentified (auto) immunity factors that act more aggressively in RN4220 in the case of SaPI2 compared to the other two SaPIs tested.

Page 2, lines 28-29: "The effect that SaPIs might have during phage infection has not been studied using spot assays. " -> It is still not clear why one would expect it to be different. My hypothesis would be that bacteria could express different receptors in liquid or solid, but this is not stated, nor a reference for an example or dynamics of infection being different between liquid and solid environments is included. Which brings me to the question (which

maybe I missed), is the effect of stl2-mediated inhibition of the phage observed also in liquid media, or only in plaques? This wouldn't change (for me) the impact of the result, but it might provide more information regarding the ecological impact of the stl2-mediated inhibition mechanism.

We would not expect the spot assays to be different compared to plaque assays. In liquid infection, there is also Stl2-mediated immunity. We were actually surprised that the phage was so severely affected by the presence of the SaPI in the spot assays. The reviewer's hypothesis is indeed very plausible.

Page 8, Lines 38-39: "may contribute to explain the widespread presence of these phage parasites in virtually any microbiome on earth" -> Although satellites have been shown to be well spread amongst different species of bacteria, as far as I am aware, that they are present in "virtually any microbiome" has not been systematically demonstrated (using, e.g., metagenomic data). Please add a reference (in the case I am missing something) or adjust the sentence accordingly.

Reference added where authors used metagenomic data.

<https://doi.org/10.1073/pnas.2212722119>

Finally, I think the authors missed my point about other satellites ACTING as (and not encoding) defense systems. When I mentioned PLE, it is because PLE itself does not have accessory genes providing defense systems, but the PLE itself IS itself the defense system, as its hijacking mechanism prevents the phage from replicating and then kills the cell. To me, this is different from the satellite-encoding defense systems (like the P4s and their systems in the Rousset et al reference), since, again, these are accessory genes, and not part of the mechanistic functions of hijacking of the satellite itself. And I believe this is what the authors are proposing here as well, by saying that SaPI2 does not encode other (accessory) defense systems, but that it is the mechanism of hijacking itself that protects against the phage. So it would be nice to put this in context with the other satellites that do the same.

We have included a sentence and a reference to put it in context: *In our view, this suggests that the phage parasite itself is used by the host as an anti-phage system, similarly to other phage parasitizing mobile elements such as the phage-inducible chromosomal-like elements, PLEs*³³.